

# Quasi-characters in $\widehat{su(2)}$ current algebra at fractional levels

**Sachin Grover⋆**

Harish-Chandra Research Institute, Chhatnag Road, Jhunsi, Allahabad, India-211019
Homi Bhabha National Institute, Training School Complex,
Anushaktinagar, Mumbai, India-400094

⋆ sachingrover@hri.res.in

## Abstract

We study the even characters of $\widehat{su(2)}$ conformal field theories (CFTs) at admissible fractional levels obtained from the difference of the highest weight characters in the unflavoured limit. We show that admissible even character vectors arise only in three special classes of admissible fractional levels which include the threshold levels, the positive half-odd integer levels, and the isolated level at -5/4. Among them, we show that the even characters of the half-odd integer levels map to the difference of characters of $\widehat{su(2)}_{4N+4}$, with $N \in \mathbb{Z}_{>0}$, although we prove that they do not correspond to rational CFTs. The isolated level characters maps to characters of two subsectors with $\widehat{so(5)}_1$ and $\widehat{su(2)}_1$ current algebras. Furthermore, for the $\widehat{su(2)}_1$ subsector of the isolated level, we introduce discrete flavour fugacities. The threshold levels saturate the admissibility bound and their even characters have previously been shown to be proportional to the unflavoured characters of integrable representations in $\widehat{su(2)}_{4N}$ CFTs, where $N \in \mathbb{Z}_{>0}$ and we reaffirm this result. Except at the three classes of fractional levels, we find special inadmissible characters called quasi-characters which are nice vector valued modular functions but with $q$-series coefficients violating positivity but not integrality.

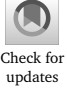

## Contents



# 1   Introduction

The conformal field theories (CFTs) with a real fractional level Kač-Moody algebra as the current algebra have been of interest for a long time. Recently, the connection of the fractional level theories with the unitary four-dimensional $\mathcal{N} = 2$ superconformal field theories (SCFTs) was brought to light [1]. They showed that the twisted-translated Schur sector of the four-dimensional $\mathcal{N} = 2$ superconformal field theories (SCFTs) when restricted to a 2-dimensional plane has a current algebra generically at fractional levels. In particular, the Schur index of the 4-dimensional SCFT is the unflavoured vacuum character of a non-unitary 2-dimensional CFT with a chiral algebra possibly at a fractional level. This result formed our initial motivation to study the unflavoured characters of the fractional level theories. In this paper, we classify the $\widehat{su(2)}$ current algebra CFTs at admissible fractional levels by imposing conformal bootstrap constraints on their unflavoured even characters.

Fractional levels in the context of $\widehat{su(2)}$ current algebra, here and from now on refer to the admissible fractional levels defined by Kač and Wakimoto [2], parametrised as $m = p/u - 2$ with $p, u \in \mathbb{Z}_{\geq 2}$ and coprime, as admissibility implies $p \geq 2$. At these levels, the highest weight modular invariant representations are finite-dimensional representations of the modular group $SL(2, \mathbb{Z})$. Since they form a finite-dimensional representation, the fractional level theories were a candidate for *rational* CFTs (RCFTs). However, it was shown in [3] that the corresponding RCFT Verlinde fusion rules [4] do not generate positive semi-definite fusion coefficients. Many CFT proposals with $\widehat{su(2)}$ current algebra at fractional levels were suggested but could not resolve all the issues plaguing the description. See for example [5–9]. The resolution of the puzzle lies in the fact that the 2-dimensional CFT with current algebra at a fractional level, not only have a finite number of highest weight modules but also an infinite number of admissible modules,[1] obtained by spectral flow automorphisms with an unbounded spectrum, and additionally, it has indecomposable modules. Thus, these theories are *logarithmic* CFTs (log CFTs) [11–19]. Contrast this situation to the theories at positive integer levels. At positive integer levels, the Wess-Zumino-Witten (WZW) models can be described as

---

[1]The notion of admissibility is extended from the notion of highest weight modules of the vertex operator algebra and the category $\mathcal{O}$ as defined in [10] to the category with relaxed-highest weight modules and its spectral flows. The definitions can be seen in [11–13].

*rational* CFTs. The integer level WZW models have a finite number of irreducible and unitary integrable highest-weight representations (primaries) of the current algebra that close under fusion, with positive semi-definite integer Verlinde fusion coefficients. Notably, the finite number of characters of the integrable representations compose the modular invariant partition function with positive semi-definite and finite integer coefficients.[2]

The character of $\widehat{su(2)}$ at level $m$ in spin-$j$ representation[3] is a vector valued modular function which is represented as

$$\chi_j(\tau,z) = Tr_{\mathcal{R}_j} q^{L_0 - c/24} \omega^{J_0^0}, \tag{1}$$

with $q = e^{2\pi i \tau}$, where $\tau \in \mathbb{H}$ and the flavour fugacity $\omega = e^{2\pi i z}$, where $\omega \in u(1)$ corresponds to the Cartan of $su(2)$ denoted by $J_0^0$. The characters (1) do not have a well-defined unflavoured limit $\omega \to 1$ for arbitrary rational spins, i.e., spins which are not integers or half-odd integers. The highest weight characters of the fractional level $\widehat{su(2)}$ theories are given by Kač-Wakimoto [2]. Although the highest weight characters are linearly independent, they do not form a basis of characters for the extended category of admissible modules, which includes the relaxed category along with the indecomposables as defined in [11,13]. Moreover, the unflavoured limit of the highest weight characters with rational spins which are not half-integers, obtained by setting the flavour fugacities to one is not a well-defined limit due to the presence of poles as we will witness in the next section. It was first observed in [22] that the difference of the characters of the highest weight primaries with equal conformal dimensions, which we will refer to as the even characters, admits a well-defined unflavoured limit or $q$-expansion. These unflavoured even characters, which are the principal subject of this paper, are poised to be the unflavoured characters of a tentative RCFT as they satisfy an (untwisted) modular linear differential equation (MLDE) with the Wronskian $l/6 = 0$ [22].[4] Usually, not all solutions of an MLDE are characters of an RCFT and a check is needed to weed out the characters which are not admissible.[5] In particular, not all solutions possess a $q$-expansion with positive integral coefficients. The procedural checks have been routinely carried out to generate admissible characters [24–29] in a constructive program to classify the RCFTs, which is still in its infancy except for the two-character theories. Besides proving a complete classification of all admissible characters for the two-component character vectors, [26] also introduced some useful applications of a class of inadmissible characters called *quasi-characters*. The quasi-characters, although nice vector valued modular functions with integer coefficients, are not admissible characters of any CFT because their $q$-expansion does not necessarily have all positive integer coefficients. As a result, we cannot interpret them as a partition sum of any CFT.[6]

The equivalence between the modules of the fractional level theory and the characters is only within the region of convergence of the characters. The highest weight characters of $\widehat{su(2)}$

---

[2]Note that a finite number of CFT primaries is insufficient to ensure its rationality (for an example see [20]). See [21] for a few definitions of rationality.

[3]Unless otherwise noted, representations in this context would mean irreducible highest weight representations.

[4]Mathur, Mukhi, and Sen (MMS) [23,24] developed a classification program for RCFTs in terms of the number of Kač-Moody characters, $r$, of the RCFT without specifying the Kač-Moody algebra. The unflavoured Kač-Moody characters satisfy an order-$r$ linear differential equation. See [25–29] for some recent applications of the MLDE approach to RCFT classification, and [30] for a review. Using this approach, most notably, MMS obtained a partial classification for the two character theories, the so-called MMS series. The two-character unitary theories with $c < 25$ were completely classified very recently in [31].

[5]We will define admissible, inadmissible and quasi-characters in the next section. The notion of an admissible character should not be confused with Kač-Wakimoto notion of admissible representations.

[6]However, [26,31,32] showed that quasi-characters have more applications. Namely, certain linear combinations of the quasi-characters can be used to build admissible characters by appropriately choosing a linear combination of two quasi-characters with the same modular $T$- transformation and appropriate modular $S$-transformation, or by a multiplication of the quasi-character by a suitable power of the modular $j$-function.

modules have a pole at $\omega = 1$ (or $z = 0$), i.e., the radius of convergence does not include the unflavoured limit [11, 13, 17]. Nevertheless, the even linear combination, which gets rid of the pole at $\omega = 1$, has a radius of convergence $1 < |\omega| < |q|^{-1}$ (or $|q| < |\omega| < 1$) for $|q| < 1$ with a well-defined unflavoured limit. The unflavoured even characters have been widely used in the literature [1, 33–36] in the context of 4-dimensional SCFTs.[7] The Schur index of a 4-dimensional SCFT is a modular object which satisfies a (twisted or untwisted) MLDE [33]. The 4-dimensional SCFTs with a $\widehat{su(2)}$ chiral algebra at admissible fractional levels will have a Schur index equal to the vacuum character, which satisfies an MLDE with $l = 0$ and transforms under modular transformations to the even characters. The modular behaviour of the Schur index is connected to the Weyl anomaly coefficient of the 4-dimensional SCFT [33].

## 1.1 Summary of results and organisation

In our study of the even characters, we show that the even unflavoured characters of $\widehat{su(2)}$ CFTs may generate a $q$-expansion with negative coefficients, i.e., seem to generically admit *quasi-characters*. Also in the cases we consider, the quasi-characters can not be made admissible characters by choosing a suitable linear combination of characters with the same modular $T$-transformations. We thus conclude that the even characters are not necessarily equivalent to modules of a CFT.

Interestingly, at certain special admissible levels labelled by $(p = 2, u = 2N + 1)$, or $(p = 2N + 3, u = 2)$, or $(p = 3, u = 4)$, where $N \in \mathbb{Z}_{\geq 0}$, we show that the even characters do not contain any quasi-characters. The even characters are admissible if they satisfy the admissibility conditions defined in section 2. To show the positivity and integrality, which are necessary conditions for the admissibility of the even characters at these special levels, we rediscover a relation between the characters of integrable representations of $\widehat{su(2)}$ at level $4N$ and the even characters of the threshold sequence $(p = 2, u = 2N + 1)$,[8] and discover new relations for the half-odd integer levels $(p = 2N + 1, u = 2)$, where $N \in \mathbb{Z}_{>0}$,[9] and in another new relation for the isolated level $(p = 3, u = 4)$ we show that there are two independent subsets of unflavoured characters: a subset of characters is related to the unflavoured $\widehat{so}(5)$ characters at level 1 and another subset to $\widehat{su(2)}$ characters at level 1. Although the positivity of the even characters is not manifest at the positive half-odd integer levels $(p = 2N+3, u = 2)$ from the analytic relations, we have checked the positivity to a sufficiently high order $\sim q^{2000}$. We use the mappings of the even characters to the unflavoured characters of integrable representations and the available modular data to establish a new correspondence between the two subsectors of $\widehat{su(2)}_{-5/4}$ log CFT, i.e., $(p = 3, u = 4)$, with $\widehat{su(2)}_1$ and $\widehat{so(5)}_1$ RCFTs, and prove that the even characters from positive half-odd integer levels do not correspond to any RCFT.

The organisation of the paper is as follows. In section 2 we set up the notation and define the admissible and quasi-characters. Then, we write the closed form expression of the flavoured characters of the $\widehat{su(2)}$ algebra, as given by Kač and Wakimoto [2]. We define the even and odd combinations of the characters [22] to extract unflavoured characters. We also define the radius of convergence of these characters which is also the radius of convergence of the characters of the integrable representations. In section 3 we study the even admissible $\widehat{su(2)}$ character vectors of the two infinite sequences $(p = 2, u = 2N + 1)$, $(p = 2N + 3, u = 2)$, where $N \in \mathbb{Z}_{>0}$ and the isolated level $(p = 3, u = 4)$. We relate the characters to integrable highest weight characters of $\widehat{su(2)}$ and $\widehat{so(5)}$ at positive integer levels through the process of unitarisation, which is based on choosing the lowest dimension primary as the unique vacuum

---

[7]Note that the unitary 4-dimensional SCFT exists only if certain unitarity bounds on the central charges of the 4-dimensional theory are satisfied (see for example the lecture notes [37]).

[8]The threshold levels saturates the Kač-Wakimoto admissibility bound on the levels.

[9]Note that $(p = 3, u = 2)$ has a quasi-character first noticed in [17].

for the unitary theory. See [36, 38–40] for some discussion. The unflavoured characters of the sequence ($p = 2N + 1, u = 2$) possess certain peculiar properties which rule them out as possible unflavoured characters of an RCFT. We will also elaborate on these properties in section 3.

We conclude the paper with a summary of our results and speculations in section 4. The appendices contain some explicit details and results. A few explicit examples of quasi-characters are given in appendix A with tabulated data on the quasi-characters in the supplementary table D. In the appendix B we explicitly write the asymptotic behaviour of the $q$-expansion, for small $q$, of the unflavoured characters for arbitrary values of $p$ and $u$. The location of the quasi-characters will be immediately clear from the rearranged $q$ expansion. The results are in complete agreement with our data on explicit $q$-expansions. In the last appendix C, we will argue that the modular $S$-matrix of the theories with quasi-characters intertwines the quasi-characters with the admissible characters. This leads to two criteria, based on the modular and conformal properties of the characters, which may rule out the possibility of a closed subsector without a quasi-character.

## 2 Highest weight characters of $\widehat{su(2)}$ at admissible levels

The flavoured characters of $\widehat{su(2)}$ chiral algebra capture the full information of the module, degeneracy for the $su(2)$ spin and conformal dimension. The unflavoured characters, which give the degeneracy at a grade ($L_0$ eigenspace) in the module are particularly useful for implementing bootstrap constraints. Apart from the examples mentioned in the introduction, another interesting example is the recent bootstrap classification of the 4-dimensional rank two SCFTs which has a Schur index satisfying a 4th-order MLDE [41].

The unflavoured characters of an RCFT are holomorphic functions in the upper half plane $\tau \in \mathbb{H}$ except for the finite number of singularities at the cusp points. This condition is called weak holomorphicity. In particular, they have the following $q$ expansion,

$$\chi_j(\tau) = q^{h_j - c/24} \sum_{i \in \mathbb{Z}_{\geq 0}} a_i^{(j)} q^i, \quad q = e^{2\pi i \tau}, \tag{2}$$

where $h_j$ is the conformal dimension or $L_0$ eigenvalue of the highest weight state and $c$ is the central charge of the tentative CFT. The coefficients $a_i^{(j)}$ need to satisfy certain admissibility conditions which we impose on the unflavoured characters. The admissibility conditions are necessarily satisfied by the characters of an RCFT. We define *admissible characters* as the unflavoured characters which satisfy the following character admissibility conditions.

1. Positivity and Integrality of the coefficients in the $q$-expansion. The coefficients in the formal power series in $q$ are interpreted as the degeneracy of states at an energy level in the corresponding module.[10]

2. The requirement of a unique vacuum state implies that the unflavoured vacuum character has the form,[11]
$$\chi_0(\tau) = q^{-c/24}(1 + a_1^{(0)} q + a_2^{(0)} q^2 + \cdots). \tag{3}$$

3. The requirement of positive multiplicities of the characters as building blocks of the

---

[10]See [42, 43] on integrality of characters which is closely linked to the representation of the modular group $SL(2, \mathbb{Z})$.

[11]The subscript '0' on the characters and modular matrices will be reserved for the vacuum representation.

physical modular invariant partition function, i.e.,

$$\mathcal{Z}(\tau, \bar{\tau}) = \sum_{i,j} \bar{\chi}_i(\bar{\tau}) M_{ij} \chi_j(\tau), \tag{4}$$

where $M_{ij} \in \mathbb{Z}_{\geq 0}$.

The unflavoured characters transform under the modular $S$-transformations as vector valued modular functions,[12]

$$\chi_j(-1/\tau) = \sum_{j'} S_{jj'} \chi_{j'}(\tau). \tag{5}$$

We can only impose positivity of the Verlinde fusion coefficients as the additional requirement on the admissible characters if the $S$-matrix in eqn (2.4) is unitary [24]. To put it a little differently, we generally do not expect the $S$ and $T$-modular transformation matrices of eqn (2.4), which act on the unflavoured characters (called the reduced $S$ and $T$-modular transformation matrices in [29]), to be compatible with the MTC structure, in particular, corresponding Verlinde fusion coefficients are not necessarily positive integers. However, a unitary $S$-matrix can be obtained from the reduced $S$-matrix following standard procedures, which is by construction compatible with the MTC structure (see [24] for more details). In particular, the unitary $S$-matrix produces positive integer Verlinde fusion coefficients,

$$N_{ijk} = \sum_{l} \frac{S_{il} S_{jl} (S^{-1})_{kl}}{S_{0l}} \in \mathbb{Z}_{\geq 0}. \tag{6}$$

The vector valued modular functions which do not satisfy the positivity of the coefficients in the $q$-expansion are called *quasi-characters*.[13] Quasi-characters routinely appear as solutions of modular linear differential equations (MLDEs), but the quasi-characters gained popularity due to their constructive usage in building admissible characters [26, 31, 32]. The characters which are not admissible are called *inadmissible characters*.

## 2.1 Kač-Wakimoto highest weight characters

The admissible levels of the $\widehat{su(2)}$ algebra were defined by Kač and Wakimoto in [2] from the modular properties of the characters of the admissible highest weight representations. Kač and Wakimoto showed that the admissible rational levels of the $\widehat{su(2)}_m$ affine Lie algebra can be parametrised in terms of two positive integers as $m = p/u - 2$ with $p, u \in \mathbb{Z}_{\geq 2}$ and coprime. The classification of all positive energy irreducible representations of the $\widehat{su(2)}$ algebra at an admissible level was done in [10] which showed that the finite number of irreducible representations (objects) from category $\mathcal{O}$ are exactly modular invariant. The set of 'admissible' representations was extended to the relaxed highest weight modules in [17]. We will restrict our attention to the highest weight modules in category $\mathcal{O}$ since they are related to integrable modules of integer level current algebra CFTs, as we will see in the remaining paper. The highest weight representations at level $m = p/u - 2$ are further parametrised in terms of two integers $1 \leq n + 1 \leq p - 1$ and $0 \leq k \leq u - 1$ as

$$2j(n,k) + 1 = n + 1 - \frac{kp}{u}, \tag{7}$$

---

[12]Strictly speaking, these are the 'reduced' $S$-matrices. The reduced $S$-matrix relates the transformation of the unflavoured characters. We will call these matrices as just $S$-matrices.

[13]The corresponding $S$-matrices also need not satisfy the positivity of the fusion coefficients.

where $j(n,k)$ is the spin of the representation. The conformal dimension of the primary with spin $j(n,k)$ is

$$h(n,k) = \frac{j(j+1)u}{p} = \frac{(nu-kp)^2}{4up} + \frac{(nu-kp)}{2p}. \tag{8}$$

The $L_0$ eigenvalue, (8) of the highest weight states remains invariant under the transformation $\sigma_0 : j \mapsto -1 - j$. This can be stated in terms of $n, k$ as

$$\bar{n} + 1 = p - (n+1), \tag{9}$$
$$\bar{k} = u - k, \quad k \neq 0, \tag{10}$$

under which $j(\bar{n}, \bar{k}) = -1 - j(n,k)$ and $h(\bar{n}, \bar{k}) = h(n,k)$.

The closed form expression for the Kač-Wakimoto character formula [2] is given in terms of

$$b_\pm(n,k) = \pm u(n+1) - kp, \quad a = pu. \tag{11}$$

Notice, $\sigma_0 : b_+ \mapsto -b_+$. The spin $j$ character is written in terms of theta functions as

$$\chi_{j(n,k)}(\tau,z) = \frac{\Theta_{b_+,a}(\tau,z/u) - \Theta_{b_-,a}(\tau,z/u)}{\Theta_{1,2}(\tau,z) - \Theta_{-1,2}(\tau,z)}, \tag{12}$$

where

$$\Theta_{b,a}(\tau,z) = \sum_{r=\mathbb{Z}+b/2a} q^{ar^2}\omega^{ar}, \tag{13}$$

where $q = e^{2\pi i\tau}$, with $\tau \in \mathbb{H}/SL(2,\mathbb{Z})$ and the flavour fugacity is $\omega = e^{2\pi iz} \in u(1)$. We can understand the radius of convergence of the character (12) by writing the denominator using the Jacobi triple product identity [17],

$$\Theta_{1,2}(\tau,\omega) - \Theta_{-1,2}(\tau,\omega) = q^{1/8}\omega^{1/2}\prod_{i=1}^{\infty}(1 - \omega^{-1}q^{i-1})(1 - q^i)(1 - \omega q^i). \tag{14}$$

The denominator has zeroes at $\omega = q^i$, where $i \in \mathbb{Z}$. The denominator can be power series expanded in $z$,

$$\Theta_{1,2}(\tau,z) - \Theta_{-1,2}(\tau,z) = 2\pi iz\eta^3(q) + \mathcal{O}(z^3). \tag{15}$$

The unflavoured limit $z \to 0$ is not well-defined due to the pole at $z = 0$. To see this, we need to expand the character in a power series of $z$ about $z = 0$. Using the series form of the $\Theta$ function,

$$\Theta_{b_+,a}(\tau,z) = \sum_{s\in\mathbb{Z}} q^{as^2 + b_+^2/4a + sb_+}\omega^{as+b_+/2}, \tag{16}$$

in the characters and then expanding it in powers of $z$ gives

$$\chi_{j(n,k)}(q,z) = \frac{1}{2\pi iz\eta^3(q)}\Big(q^{b_+^2/4a}\sum_{s\in\mathbb{Z}} q^{as^2+sb_+} - q^{b_-^2/4a}\sum_{s\in\mathbb{Z}} q^{as^2+sb_-}\Big) + \mathcal{O}(z)$$
$$+ \frac{1}{u\eta^3(q)}\Big(q^{b_+^2/4a}\sum_{s\in\mathbb{Z}} q^{as^2+sb_+}(as+b_+/2) - q^{b_-^2/4a}\sum_{s\in\mathbb{Z}} q^{as^2+sb_-}(as+b_-/2)\Big). \tag{17}$$

The pole represents the infinite number of states in each grade generated by the action of $su(2)$ currents which is due to the fractional $su(2)$ weights. For $k = 0$, $b = b_+ = -b_-$ the character takes the form, as $z \to 0$,

$$\chi_{j(n,0)}(q) = \frac{1}{\eta^3(q)} q^{(n+1)^2/4(m+2)} \sum_{s\in\mathbb{Z}} q^{us((n+1)+us(m+2))}(n+1+2us(m+2)). \tag{18}$$

All of the characters labelled by $(n,0)$ are admissible characters. They have half integer spins and are denoted by $\mathcal{L}_j$. Their characters do not have singularity in the $z \to 0$ limit and in fact, the characters are analogous to the integrable highest weight characters of $\widehat{su(2)}_M$ where $M \in \mathbb{N}$ [44],

$$\chi_l^{(M)}(q) = \frac{q^{(l+1)^2/4(M+2)}}{\eta^3(q)} \sum_n \left(l+1+2n(M+2)\right)q^{n(l+1+(M+2)n)}, \tag{19}$$

where we have used the notation $\chi_l^{(M)}$ to denote the character of spin $l/2$ representation of $\widehat{su(2)}$ at level $M$. The characters have a $q$-expansion (19) which corresponds directly to the energy eigenspace degeneracies of the corresponding CFT modules.

## 2.2 Even and odd characters

For the characters with $k \neq 0$, we need to define even ($\chi^+$) and odd ($\chi^-$) characters which are the following linear combinations of the highest weight characters [22],

$$\chi_{j(n,k)}^{\pm}(q,\omega) = \chi_{j(n,k)}(q,\omega) \mp \chi_{j(\bar{n},\bar{k})}(q,\omega). \tag{20}$$

The odd characters are related to the Virasoro minimal model characters [22], thus satisfying admissibility conditions (2), (3) as well as positivity and integrality of Verlinde fusion coefficients. The odd characters are the characters of the indecomposable admissible modules, and the indecomposable modules are proportional to the characters of the irreducible Virasoro modules [13].

We turn our attention to the even characters. The even combination of characters does not possess any singularity in the $\omega \to 1$, which we demonstrate below, first for an example $\widehat{su(2)}_{-1/2}$ theory to convince the reader before we move to the general explanation.

**Example.** We begin by the $\widehat{su(2)}_{-1/2}$ example parametrised with integers $p = 3$ and $u = 2$. Apart from the vacuum representation, we have two more highest weight representations with spins $j(0,1) = -3/4$ and $j(1,1) = -1/4$. The character $\chi_{j(0,1)=-3/4}$ obtained by using (12), is

$$\chi_{-3/4}(q,\omega) = q^{-1/12}\omega^{-3/4}\frac{\sum_{s\in\mathbb{Z}} q^{6s^2-s}\omega^{3s} - \sum_{s\in\mathbb{Z}} q^{6s^2+1-5s}\omega^{3s-1}}{\sum_{s\in\mathbb{Z}} q^{2s^2+s}\omega^{2s} - \sum_{s\in\mathbb{Z}} q^{2s^2-s}\omega^{2s-1}}.$$

Performing the polynomial division, we have an infinite series expansion,

$$\chi_{-3/4}(q,\omega) = -q^{-1/12}\omega^{1/4}\left(1+\omega+\omega^2+\cdots\right) - q^{11/12}\omega^{1/4}\left(1+2\omega+2\omega^2+\cdots\right) + \mathcal{O}(q^{23/12}). \tag{21}$$

The $\omega$ series can be resummed in different radii of convergence to obtain a $q$-series with $\omega$ dependent coefficients,[14]

$$\chi_{-3/4}(q,|\omega|<1) = -q^{-1/12}\frac{\omega^{1/4}}{1-\omega}\left(1+q(1+\omega)+\mathcal{O}(q^2)\right), \tag{22}$$

$$\chi_{-3/4}(q,|\omega|>1) = q^{-1/12}\frac{\omega^{3/4}}{1-\omega}\left(1+q(1+\frac{1}{\omega})+\mathcal{O}(q^2)\right). \tag{23}$$

---

[14]Note that we are interested in the regions $|q| < |\omega| < 1$ or $1 < |\omega| < |q|^{-1}$ where $|q| < 1$.

These expansions are, of course, related by the transformation $\omega \to 1/\omega$. We note that the expansion (23) can be recognised with positive coefficients. Similarly, the other character with spin $j = -1/4$ has the expansions

$$\chi_{-1/4}(q, |\omega| < 1) = -q^{-1/12} \frac{\omega^{3/4}}{1 - \omega} \left( 1 + q \left( \frac{1}{\omega} + 1 \right) + \mathcal{O}(q^2) \right), \tag{24}$$

$$\chi_{-1/4}(q, |\omega| > 1) = q^{-1/12} \frac{\omega^{1/4}}{1 - \omega} \left( 1 + q (1 + \omega) + \mathcal{O}(q^2) \right). \tag{25}$$

From these characters, we can extract an even character,

$$\chi_{-1/4}^{+}(q, |\omega| > 1) = \chi_{-1/4}(q, |\omega| > 1) - \chi_{-3/4}(q, |\omega| > 1) \tag{26}$$

$$= q^{-1/12} \frac{\omega^{1/4}(1 - \omega^{1/2})}{1 - \omega} \left( 1 - q \left( \frac{1}{\sqrt{\omega}} + \sqrt{\omega} \right) + \mathcal{O}(q^2) \right), \tag{27}$$

and $\chi_{-1/4}^{+}(q, |\omega| < 1)$ obtained by the applying the rule $\omega \to 1/\omega$ to (26). The unflavoured limit on the real axis $\omega \to 1$ of eqn (26) is well-defined,

$$\chi_{-1/4}^{+}(q, \omega \to 1) = q^{-1/12} \frac{1}{2} \left( 1 - 2q + \mathcal{O}(q^2) \right), \tag{28}$$

defined in the region $|q| < 1$. We can scale the character by $u = 2$ to obtain an integral $q$-expansion which is the *unflavoured even character* or simply as even character, whenever the meaning is evident. In fact, for $\widehat{su(2)}_{-1/2}$ we only have a single linearly independent unflavoured even character since

$$\chi_{-1/4}^{+}(q, |\omega| > 1) + \chi_{-3/4}^{+}(q, |\omega| < 1) = 0,$$
$$\chi_{-1/4}^{+}(q, |\omega| < 1) + \chi_{-3/4}^{+}(q, |\omega| > 1) = 0. \tag{29}$$

The above calculation illustrates how to regularise the divergent infinite series in eqn (23) to the unflavoured limit (28).

Next, we generalise the above derivation to an arbitrary Kač-Wakimoto admissible level $m = p/u - 2$. The character of the irreducible highest weight representation with spin $j(n, k)$ is[15]

$$\chi_{j(n,k)}(q, \omega) = \frac{\sum_{s \in \mathbb{Z}} \omega^j \left( q^{as^2 + b_+^2/4a + sb_+} \omega^{as/u} - q^{as^2 + b_-^2/4a + sb_-} \omega^{as/u - (n+1)} \right)}{q^{1/8} \prod_{i=1}^{\infty} (1 - \omega^{-1} q^{i-1})(1 - q^i)(1 - \omega q^i)}, \tag{30}$$

with the character in the conjugate representation can be brought to the form

$$\chi_{j(\bar{n},\bar{k})}(q, \omega) = \frac{\sum_{s \in \mathbb{Z}} \omega^{-j-1} \left( q^{as^2 + b_+^2/4a + sb_+} \omega^{-as/u} - q^{as^2 + b_-^2/4a + sb_-} \omega^{-as/u + (n+1)} \right)}{q^{1/8} \prod_{i=1}^{\infty} (1 - \omega^{-1} q^{i-1})(1 - q^i)(1 - \omega q^i)}. \tag{31}$$

The even character is just the subtraction of the above two characters, (30) and (31).

$$\chi_{j(\bar{n},\bar{k})}^{+}(q, \omega) = \frac{\sum_{s \in \mathbb{Z}} q^{as^2 + b_+^2/4a + sb_+} \left( \omega^{as/u + j + 1} - \omega^{-as/u - j} \right)}{q^{1/8}(\omega - 1)(1 - q)(1 - q\omega) \prod_{i=2}^{\infty} (1 - \omega^{-1} q^{i-1})(1 - q^i)(1 - \omega q^i)}$$

$$- \frac{\sum_{s \in \mathbb{Z}} q^{as^2 + b_-^2/4a + sb_-} \left( \omega^{as/u + j + 1 - (n+1)} - \omega^{-as/u - j + (n+1)} \right)}{q^{1/8}(\omega - 1)(1 - q)(1 - q\omega) \prod_{i=2}^{\infty} (1 - \omega^{-1} q^{i-1})(1 - q^i)(1 - \omega q^i)}. \tag{32}$$

---

[15]We identify $j(n,k) \equiv j$, $b_+(n,k) = b_+$, $b_-(n,k) = b_-$ here.

The unflavoured limit $\omega \to 1$ is non-trivial for the factors

$$\frac{\omega^{as/u+j+1} - \omega^{-as/u-j}}{\omega - 1} \quad \text{and} \quad \frac{\omega^{as/u+j+1-(n+1)} - \omega^{-as/u-j+(n+1)}}{\omega - 1} \tag{33}$$

of the even character (32). The rest of the character is regular near the point $\omega = 1$ in the region $|q| < 1$. We can obtain an infinite series by expanding the denominator in the region $|\omega| > 1$ (or equivalently $|\omega| < 1$). The limit $\omega \to 1$ is well-defined for all such characters only on the real $\omega$ line, which yields the unflavoured limit

$$\chi^{+}_{j(n,k)}(q) = \frac{1}{u\eta^3(q)} \Big( q^{b_+^2/4a} \sum_{s\in\mathbb{Z}} q^{as^2+sb_+}(2as + b_+) - q^{b_-^2/4a} \sum_{s\in\mathbb{Z}} q^{as^2+sb_-}(2as + b_-) \Big). \tag{34}$$

Note that the even character (34) is precisely the $z$ independent coefficient in the infinite series (17) multiplied by a factor of 2. Also, note that

$$\chi^{+}_{j(\bar{n},\bar{k})}(q,\omega) = \chi_{j(\bar{n},\bar{k})}(q,\omega) - \chi_{j(n,k)}(q,\omega) = -\chi^{+}_{j(n,k)}(q,\omega). \tag{35}$$

Due to this relation, we have a pair of identical characters up to a negative sign in the set of even combinations of characters.

Let us now discuss the modular properties of the even characters. While the action of modular transformation $T$ on the characters is by a left multiplication by a diagonal matrix

$$T_{nk,n'k'} = e^{2\pi i h(n,k)} \delta_{nk,n'k'}, \tag{36}$$

the characters transform non-trivially under the modular $S$-transformation. The $S$-matrix components [22] are

$$S_{nk,n'k'} = \sqrt{\frac{2}{pu}}(-1)^{k'(n+1)+k(n'+1)} e^{-i\pi kk'p/u} \sin\left(\pi \frac{(n+1)(n'+1)u}{p}\right). \tag{37}$$

The modular $S$-transformations of the even characters can then be written in the following manner,

$$\chi^{+}_{j(n,k\neq0)}\left(-\frac{1}{\tau}, \frac{z}{\tau}\right) = \sum_{n',k'\neq0} 2S^{+}_{nk,n'k'}\chi^{+}_{j(n',k')}(\tau,z) + \sum_{n'} 2S_{nk,n'0}\chi_{j(n',0)}(\tau,z), \tag{38}$$

and

$$\chi_{j(n,0)}\left(-\frac{1}{\tau}, \frac{z}{\tau}\right) = \sum_{n',k'\neq0} S_{n0,n'k'}\chi^{+}_{j(n',k')}(\tau,z) + \sum_{n'} S_{n0,n'0}\chi_{j(n',0)}(\tau,z), \tag{39}$$

where the sum runs over the labels $(n', k')$ of even characters. The components of the modular $S^{+}$-transformation matrix are

$$S^{+}_{nk,n'k'} = \frac{1}{2}\Big( S_{nk,n'k'} - S_{\bar{n}\bar{k},n'k'} \Big). \tag{40}$$

Thus we see that the even characters form a closed modular invariant set under the modular $S$ and $T$ transformations which we will refer to as the even sector. As mentioned above we will work with the even characters and in particular we will focus on the unflavoured characters, *i.e.*, they are functions of the modular parameter $\tau$ only.

### 2.3 Unflavoured even characters

In this subsection, we will analyse the $q$-series expansion of even characters $\chi^+$ in the $\omega \to 1$ limit. Since we have eliminated the pole at $\omega = q^0 = 1$, the even characters have a well-defined $q$-expansion in the region $|q| < 1$ in the limit $\omega \to 1$, with finite integer coefficients which are also positive for specific cases. Let us now focus on the unflavoured limit of the even characters.

Due to the $\sigma_0$ symmetry, even characters always appear in pairs, one with $b_+ < 0$ and the other with $b_+ > 0$. We, therefore, restrict ourselves to $b_+ < 0$ in this subsection.[16]

The $q$-expansion of even characters is unique in the well defined limit $z \to 0$ and is given by

$$\chi^+_{j(n,k)}(q) = \frac{q^{b_+^2/4a-1/8}}{\eta^3(q)}\Big[\sum_{s\in\mathbb{Z}} 2sp\, q^{s^2a+sb_+} + \frac{b_+}{u}\sum_{s\in\mathbb{Z}} q^{s^2a+sb_+}\Big]$$
$$- \frac{q^{b_-^2/4a-1/8}}{\eta^3(q)}\Big[\sum_{s\in\mathbb{Z}} 2sp\, q^{s^2a+sb_-} + \frac{b_-}{u}\sum_{s\in\mathbb{Z}} q^{s^2a+sb_-}\Big]. \tag{41}$$

The exponent of the leading term in the $q$-series in the $\tau \to i\infty$ limit can be identified as

$$b_+^2/4a - 1/8 = -c(p,u)/24 + h(n,k), \tag{42}$$

where $c(p,u)$ is the Virasoro central charge and $h(n,k)$ is the conformal dimension. The set of even characters needs to be multiplied by $u$ which ensures that the coefficients are integers and the modular transformations remains invariant, however, as we will see, positivity of the coefficients is not guaranteed.

The terms can now be rearranged in an ascending order of powers of $q$. There are two orderings of the expansion based on the values of $b_-$. That is, for $|b_-| < a$, and $b_+ < 0$ we have

$$\chi^+_{j(n,k)}(\tau) = \frac{b_+}{u\eta^3(q)}q^{b_+^2/4a-1/8}\Bigg[1 - \frac{b_-}{b_+}q^{k(n+1)} - \sum_{s\in\mathbb{N}}q^{k(n+1)+s^2a+sb_-}\left(\frac{2usp}{b_+} + \frac{b_-}{b_+}\right)$$
$$+ \sum_{s\in\mathbb{N}}q^{s^2a+sb_+}\left(1 + \frac{2usp}{b_+}\right) + \sum_{s\in\mathbb{N}}q^{s^2a-sb_+}\left(1 - \frac{2usp}{b_+}\right)$$
$$+ \sum_{s\in\mathbb{N}}q^{k(n+1)+s^2a-sb_-}\left(\frac{2usp}{b_+} - \frac{b_-}{b_+}\right)\Bigg], \tag{43}$$

or, for $|b_-| > a$, and $b_+ < 0$ we have

$$\chi^+_{j(n,k)}(\tau) = \frac{b_+}{u\eta^3(q)}q^{b_+^2/4a-1/8}\Bigg[1 + \left(-\frac{2up}{b_+} - \frac{b_-}{b_+}\right)q^{k(n+1)+a+b_-} - \frac{b_-}{b_+}q^{k(n+1)}$$
$$- \sum_{s-1\in\mathbb{N}}q^{k(n+1)+s^2a+sb_-}\left(\frac{2usp}{b_+} + \frac{b_-}{b_+}\right) + \sum_{s\in\mathbb{N}}q^{s^2a+sb_+}\left(1 + \frac{2usp}{b_+}\right)$$
$$+ \sum_{s\in\mathbb{N}}q^{s^2a-sb_+}\left(1 - \frac{2usp}{b_+}\right) + \sum_{s\in\mathbb{N}}q^{k(n+1)+s^2a-sb_-}\left(\frac{2usp}{b_+} - \frac{b_-}{b_+}\right)\Bigg]. \tag{44}$$

This rearrangement paves the way for a hierarchical arrangement of the powers of $q$. This is shown in eq. (B.1) for the arrangement with $|b_-| < a$. The asymptotic expansion makes the coefficients of the $q$-series up to $q^{4a+2b_-+k(n+1)}$ appear manifestly as a function of $(n,k)$

---

[16]We will not restrict ourselves to this condition later and will choose the even sector as per convenience.

for a level parametrised as $(p, u)$. To investigate if the character is a quasi-character,[17] we look for a sign flip of the coefficients in the $q$-expansion. From these $q$-expansions we have found that there are quasi-character(s) at every level except the levels $(p = 2, u = 2N + 1)$, $(p = 2N + 3, u = 2)$, where $N \in \mathbb{Z}_{>0}$ and $(p = 3, u = 4)$.

# 3 Admissible levels with admissible even character vector

In this section, we will focus on the even characters of the $\widehat{su(2)}$ algebra at the special fractional levels which only have admissible even characters. If we look at the $q$-series expansion, (43) or (44), we see that quasi-characters are ubiquitous except at very special points in the space of admissible even characters. These special levels correspond to $(p = 2, u = 2N + 1)$, $(p = 2N + 3, u = 2)$, where $N \in \mathbb{Z}_{>0}$ and an isolated point at $(p = 3, u = 4)$. We show the positivity of the even characters by relating them to characters of integrable representations of a current algebra at an integer level. Thus, we discover two new maps from the even unflavoured characters of fractional level $\widehat{su(2)}$ current algebra log CFTs to the characters of integer level $\widehat{su(2)}$ and $\widehat{so(5)}$ current algebra RCFTs. We will also present examples of the two infinite sequences in the subsections 3.1 and 3.3. We will focus on these special levels in this section, but before we discuss that, a comment on the appearance of quasi-characters in other cases is in order.

It is obvious from the $q$-expansions of the character formula at these levels that, except at the special levels mentioned above, we always find one quasi-character with labels $(n = 0, k = \lceil (u/p) \rceil)$, where $\lceil x \rceil$ denotes the ceiling function of $x$. It is easy to verify this from the small $q$ expansion in eq. (B.1). Furthermore, for higher $k$ and $n$ values, one may find additional quasi-character(s). While we have studied the $q$-expansions for a large but finite set of $p$ and $u$ values, the existence of a quasi-character for the labels $(n = 0, k = \lceil (u/p) \rceil)$ guarantees that every $\widehat{su(2)}$ theory at fractional levels, except for the special ones, have at least one quasi-character. To get a better picture of this pattern, we present some illustrative examples in the appendix A. In appendix C we look at the fractional levels with quasi-characters in their even sector.

## 3.1 Threshold levels

Among the special levels mentioned at the beginning of this section, the set with $p = 2$ saturates the admissibility bound. For this reason, the $\widehat{su(2)}$ levels are called the threshold levels. They are also called boundary levels for the same reason. The term boundary levels was first used in [45]. Since $p = 2$, $n = 0$ is the only possible value. Due to the condition $\gcd(p, u) = 1$, $u = 2N + 1$, therefore $k = 0, \cdots, 2N$ which we restrict to $0 \le k \le N$ due to the $\sigma_0$ transformations (35). The central charge for these theories is

$$c = -6N. \tag{45}$$

To satisfy the integrality condition, the even characters are scaled by $u = 2N + 1$. For simplicity, we will call $\chi_j^+(n, k)(q)$ from now on to denote the even characters with integer coefficients, that is, all the even characters are multiplied by $u$. We need to relabel the highest weight representations such that the lowest dimensional primary is identified with the vacuum of the tentative unitary RCFT. This procedure, which we will call unitarisation in anticipation that the relabelling produces a unitary RCFT,[18] chooses the correct vacuum character of the

---

[17]After suitable rescaling by the integer $u$.

[18]At the very least, the conformal dimensions and central charge are positive after relabelling.

form given in eqn. (3). If the unitarisation procedure gives a representation theory of a known current algebra RCFT then it automatically guarantees that all the characters are admissible.

Having set up the procedure and the expectations, we illustrate this for the threshold levels. The lowest dimensional primary in the set of highest weight representations is given by

$$h_{min} = h_N = -\frac{N}{2}\left(\frac{N+1}{2N+1}\right). \tag{46}$$

We identify this representation with the vacuum representation of the unitarised theory. The character of this vacuum representation has its leading coefficient $b_+(N) = 1$, hence it is non-degenerate. The central charge of unitarised CFT is

$$c_U = c - 24h_{min} = \frac{3(4N)}{4N+2}, \tag{47}$$

which is the central charge of $\widehat{su(2)}_{4N}$ CFT. The spectrum of conformal dimensions in this theory is shifted by $h_{min}$,

$$h_k^U = \frac{(N-k)(N-k+1)}{4N+2}. \tag{48}$$

The spins are labelled as $l/2 = N - k$.

The unitarisation procedure for the threshold cases provides a map to the D-type characters of $\widehat{su(2)}_{4N}$ theories with $N \in \mathbb{Z}_{>0}$ [22].

The unflavoured even characters are equal to the unflavoured characters of the D-type modular invariants of $\widehat{su(2)}$ theory at level $4N$. This can be easily shown by manipulating the numerators of the characters, the denominator factor is independent of the level and does not play any role in the manipulation. Instead of using the label $k$, we choose the spin of the representation $l$ of the related integrable representation,

$$\chi_{j(0,k)}^+(q) = \frac{2(2N+1)}{\eta^3(q)} q^{(l+1)^2/4(4N+2)} \sum_s \left[ q^{s^2(4N+2)+s(l+1)}\left(2s + \frac{l+1}{4N+2}\right)\right.$$
$$\left. - q^{k+s^2(4N+2)+s(-4N+l-1)}\left(2s - \frac{4N-l+1}{4N+2}\right)\right]$$
$$= \chi_l^{(4N)}(q) + \chi_{4N-l}^{(4N)}(q), \tag{49}$$

and similarly

$$\chi_{j(0,0)} = \chi_{2N}^{(4N)}. \tag{50}$$

This establishes the positivity of the characters of threshold cases as well. The final expression of the characters can be compared with eq. (19), with the identification $4N = M$.

The flavoured characters do not match for arbitrary fugacities, $\omega \in u(1)$, but the vacuum character of the even sector with $u = 2N + 1$ match with the corresponding $\widehat{su(2)}_{4N}$ characters with spin $j = N$ at discrete fugacities [36],

$$\chi_N^{(4N)}(q, \omega) = \chi_{2N}^{su(2)}(\omega)\chi_0^+(q, \omega), \quad \omega = e^{\pi i l/(N+1)}, \tag{51}$$

where $\chi_{2N}^{su(2)}(\omega) = \sum_{i=-N}^{i=N} \omega^i$ is the finite $su(2)$ character in spin $N$ representation, and $0 \le l \le 2N + 1$. When $l = 0$ the (51) reduces to the unflavoured limit (49). The consequences of (51) for the 4-dimensional SCFT are discussed in [36] which we will not pursue here. In retrospect, we should have been confident of the threshold cases to produce admissible characters due to the relations in section 5.2 of [33]. We will demonstrate a similar

Table 1: The primaries in the even sector of $\widehat{su(2)}$ current algebra at level $m = -5/4$ are labelled by $(n,k)$. The spins of the primaries are labelled by $j(n,k)$ and conformal dimensions by $h(n,k)$.

| $(n,k)$ | $j(n,k)$ | $h(n,k)$ |
|---------|----------|----------|
| (0,0)   | 0        | 0        |
| (1,0)   | 1/2      | 1        |
| (0,1)   | -3/8     | -5/16    |
| (1,1)   | 1/8      | 3/16     |
| (1,2)   | -3/4     | -1/4     |

computation in the next subsection to flavour a subsector of characters at the isolated level $m = -5/4$ to the characters of $\widehat{su(2)}_1$.

The modular invariant partition function composed of the even characters is a D-type modular invariant of the A-D-E classification [46] when written in terms of characters in integrable representation, as mentioned above.

$$\mathcal{Z}^{(4N)} = 2|\chi_{2N}^{(4N)}|^2 + \sum_{l=0}^{N-1} |\chi_{2l}^{(4N)} + \chi_{4N-2l}^{(4N)}|^2. \tag{52}$$

Let us demonstrate the results through an example of $\widehat{su(2)}_{-4/3}$ theory, where we have the A-type modular invariant partition function given by[19]

$$\mathcal{Z}_{-4/3} = |\chi_{j(0,0)}|^2 + \frac{1}{2}|\chi_{j(0,1)}^+|^2, \tag{53}$$

where, $j(n,k)$ represents the spin of the highest weight state. This partition function does not have integer coefficients but we can write another modular invariant partition function,

$$2\mathcal{Z}_{-4/3} = 2|\chi_{j(0,0)}|^2 + |\chi_{j(0,1)}^+|^2, \tag{54}$$

which has a $q$-expansion with integer coefficients. We can use the relations given in eq (49) to write the partition function as the D-type modular invariant of $\widehat{su(2)}_4$. The correspondence also holds at the level of modular $S$ and $T$ matrices and consequently the fusion algebra.

## 3.2 Curious case of the isolated level $m = -5/4$ ($p = 3, u = 4$)

In this subsection, we will look at the isolated case of $m = -5/4$. This is a particularly interesting case, firstly because it is a single entry and not an infinite sequence. The labels $(n,k)$, spins $j(n,k)$, and the conformal dimensions $h(n,k)$ of the even sector are listed in table 1. The central charge of this theory is $c = -5$.

Corresponding modular $S$-matrix can be read off from eq.(37) and eq.(40),

$$\begin{pmatrix} \chi_{j(0,0)}(-1/\tau) \\ \chi_{j(1,0)}(-1/\tau) \\ \chi_{j(0,1)}^+(-1/\tau) \\ \chi_{j(1,1)}^+(-1/\tau) \\ \chi_{j(1,2)}^+(-1/\tau) \end{pmatrix} = \frac{1}{2\sqrt{2}} \begin{pmatrix} -1 & 1 & 1 & -1 & 1 \\ 1 & -1 & 1 & -1 & -1 \\ 2 & 2 & \sqrt{2} & \sqrt{2} & 0 \\ -2 & -2 & \sqrt{2} & \sqrt{2} & 0 \\ 2 & -2 & 0 & 0 & 2 \end{pmatrix} \begin{pmatrix} \chi_{j(0,0)}(\tau) \\ \chi_{j(1,0)}(\tau) \\ \chi_{j(0,1)}^+(\tau) \\ \chi_{j(1,1)}^+(\tau) \\ \chi_{j(1,2)}^+(\tau) \end{pmatrix}. \tag{55}$$

---

[19]Note that this is not the actual modular partition function of the $\widehat{su(2)}_{-4/3}$ log CFT since there are infinite number of modules contributing the same character to the partition function.

Table 2: The unitarised value of the conformal dimension associated to the primaries in the even sector of $\widehat{su(2)}$ current algebra at level $m = -5/4$ denoted by $h_U(n, k)$.

| $(n, k)$ | $h_U(n, k)$ |
|----------|-------------|
| $(0,0)$  | $5/16$      |
| $(1,0)$  | $21/16$     |
| $(0,1)$  | $0$         |
| $(1,1)$  | $1/2$       |
| $(1,2)$  | $1/16$      |

We can construct an A-type modular invariant partition function using only the even characters,

$$\mathcal{Z}(\tau, \bar{\tau}) = \sum_{i,j \in \Gamma} \bar{\chi}_i M_{ij} \chi_j, \tag{56}$$

where we have already rescaled the characters by a factor $u = 4$ so that all coefficients in the $q$-expansion are integers. The scaling forces us to choose a different vacuum as we had seen for the threshold levels. To further the reason, note that the $S$-matrix is not unitary but $S^\dagger M S = M$, where $M = \mathrm{diag}(1, 1, \frac{1}{2}, \frac{1}{2}, \frac{1}{2})$. The factors $1/2$ imply that we need to rescale the $\chi_{n,0}$ characters by 2 since it is the smallest integer which makes the coefficients integers. Hence the physical modular invariant is

$$\mathcal{Z}(\tau, \bar{\tau}) = 2|\chi_{j(0,0)}|^2 + 2|\chi_{j(1,0)}|^2 + |\chi^+_{j(0,1)}|^2 + |\chi^+_{j(1,1)}|^2 + |\chi^+_{j(1,2)}|^2. \tag{57}$$

This implies that the choice of the vacuum for the non-unitary theory is not correct anymore and we are forced to consider a different vacuum. We can choose the lowest dimensional primary as the vacuum character. In this example, it corresponds to the choice of character $\chi^+_{j(0,1)}$ to belong to the vacuum module. The resulting unitarised theory has non-negative conformal dimensions and central charge.

The RCFT description is in terms of the unitarised theory, where the primary with labels $(0, 1)$ is identified with the vacuum. Table 2 lists the unitarised conformal dimensions. The unitarised theory has the central charge

$$c_U = c - 24 h_{min} = -5 + 24(5/16) = 5/2. \tag{58}$$

The primary field with conformal dimension $1/2$ indicates the presence of a fermion. Combining this with the fact that the central charge $c_U = 5/2$ indicates that this theory has a free field representation in terms of 5 fermions which form a representation of the $\widehat{so}(5)_1$ current algebra. Note that the conformal dimensions of $(n = 0, k = 0)$ and $(n = 1, k = 0)$ differ by an integer which may give an impression that this theory can be a logarithmic CFT, however the sum of the two characters is required to build characters of the unitarised theory. Also, log CFTs are always non-unitary and there is no unitary counterpart of it. A straightforward analysis rules out the $A$-type modular invariant, we can look for a $D$-type modular invariant to accommodate this spectrum since we have the characters whose conformal dimensions are separated by an integer which can be combined to build the non-diagonal partition function. The non-diagonal modular invariant partition function has the form

$$\mathcal{Z}(\tau, \bar{\tau}) = |\chi_{j(0,1)}|^2 + |\chi_{j(0,0)} + \chi_{j(1,0)}|^2 + |\chi_{j(1,1)}|^2. \tag{59}$$

Since the vacuum character corresponds to $\chi^+_{j(0,1)}$, which has a unit leading coefficient in its $q$-expansion, ensures that the vacuum is unique for this theory. The modular $S$-transformation

of the characters is given by

$$
\begin{pmatrix} \chi^+_{j(0,1)}(-1/\tau) \\ \chi_{j(1,0)}(-1/\tau) + \chi_{j(0,0)}(-1/\tau) \\ \chi^+_{j(1,1)}(-1/\tau) \end{pmatrix} = \begin{pmatrix} 1/2 & 1/\sqrt{2} & 1/2 \\ 1/\sqrt{2} & 0 & -1/\sqrt{2} \\ 1/2 & -1/\sqrt{2} & 1/2 \end{pmatrix} \begin{pmatrix} \chi^+_{j(0,1)}(\tau) \\ \chi_{j(1,0)}(\tau) + \chi_{j(0,0)}(\tau) \\ \chi^+_{j(1,1)}(\tau) \end{pmatrix}.
\tag{60}
$$

Note that the above $S$-matrix is now orthogonal $S^T S = \mathbb{I}$ (and therefore unitary). Since we have distinct modules in the $\widehat{so(2r+1)}$ WZW models corresponding to distinct unflavoured characters, we can demand positivity of fusion rules as mentioned in section 2. We compute the Verlinde Fusion rules from the reduced $S$-matrix (60) using the Verlinde formula [4]. The fusion rules can be conveniently written in the form of the fusion matrices $(N_i)^j_k$,

$$
N_{(0,1)} = \begin{pmatrix} 1 & 0 & 0 \\ 0 & 1 & 0 \\ 0 & 0 & 1 \end{pmatrix}, \quad N_{(0,0)+(1,0)} = \begin{pmatrix} 0 & 1 & 0 \\ 1 & 0 & 1 \\ 0 & 1 & 0 \end{pmatrix}, \quad N_{(1,1)} = \begin{pmatrix} 0 & 0 & 1 \\ 0 & 1 & 0 \\ 1 & 0 & 0 \end{pmatrix}, \tag{61}
$$

where the fusion matrices are written in the basis of the vector in (60). The resulting unitary CFT is the $\widehat{so}(5)_1$ WZW model which has central charge $c = 5/2$. It can also be described in terms of 5 copies of the Ising Model. We can identify the primaries in the following manner,

$$
\begin{aligned}
(0,1) &\equiv \mathbb{I}, \\
(0,0) + (1,0) &\equiv \sigma, \\
(1,1) &\equiv \epsilon.
\end{aligned}
\tag{62}
$$

With this identification these primaries have same fusion rules as the Ising model,

$$
\begin{aligned}
\epsilon \times \epsilon &= \mathbb{I}, \\
\sigma \times \sigma &= \mathbb{I} + \epsilon, \\
\sigma \times \epsilon &= \sigma.
\end{aligned}
\tag{63}
$$

We can explicitly show that these characters are identical to the characters of $\widehat{so}(5)_1$.

$$
\chi_{j(0,1)} = \chi_{\hat{\omega}_0} = \frac{1}{2} \frac{\theta_3^{5/2} + \theta_4^{5/2}}{\eta^{5/2}}, \tag{64}
$$

$$
\chi_{j(1,1)} = \chi_{\hat{\omega}_1} = \frac{1}{2} \frac{\theta_3^{5/2} - \theta_4^{5/2}}{\eta^{5/2}}, \tag{65}
$$

$$
\chi_{j(0,0)} + \chi_{j(1,0)} = \chi_{\hat{\omega}_2} = \frac{1}{\sqrt{2}} \left( \frac{\theta_2}{\eta} \right)^{5/2}. \tag{66}
$$

Recall, we are looking only at the even characters, it is worth pointing out that the odd combination of characters for this level gives us the characters of the Ising model.

There is an interesting alternative to the choice of the sub-sector, with the choice of the representation $(1,2)$ as the vacuum representation and taking an orthogonal non-diagonal combination of $(0,0)$ and $(1,0)$, namely, $\chi_{j(0,0)} - \chi_{j(1,0)}$.[20] This set forms a modular invariant combination with the central charge $c = 1$ and corresponds to $\widehat{su(2)}_1$ WZW model. The spectrum of conformal dimensions is $h_U(1,2) = 0$ and $h_U((0,0) - (1,0)) = 1/4$. The explicit

---

[20] The decomposition will be manifest below when we write the block-diagonal form of the modular $S$-matrix.

equivalence of the characters is given below.[21]

$$\chi^{+}_{j(1,2)}(q) = \frac{2q^{1/12}}{\eta^3(q)} \sum_s \left[ q^{12s^2+2s}\left(6s+\frac{1}{2}\right) - q^{12s^2-14s+4}\left(6s-\frac{7}{2}\right) \right]$$

$$= \frac{q^{1/12}}{\eta^3(q)} \sum_s (1+6s)q^{s+3s^2} = \chi_0^{(1)}(q). \tag{67}$$

Similarly,

$$\chi_{j(0,0)}(q) - \chi_{j(1,0)}(q) = \frac{2q^{1/3}}{\eta^3(q)} \sum_s \left[ q^{12s^2+4s}(6s+1) - q^{12s^2+8s+1}(6s+2) \right]$$

$$= \frac{q^{1/3}}{\eta^3(q)} \sum_s (2+6s)q^{3s^2+2s} = \chi_1^{(1)}(q). \tag{68}$$

This identification guarantees that all coefficients in the $q$-expansion are positive. These relations complete the proof of the positivity of the characters in the representation of $(p=3, u=4)$.

The modular $S$-matrix for this two character subsector is

$$S = \frac{1}{\sqrt{2}}\begin{pmatrix} 1 & 1 \\ 1 & -1 \end{pmatrix}, \tag{69}$$

where the first row corresponds to the vacuum character $\chi_0^{(1)}(q)$. The $S$-matrix is again unitary and hence the positivity of Verlinde fusion rules can be imposed on the characters. However, fusion rules and the modular $S$ and $T$ matrix alone do not characterise the RCFT. It is straightforward to flavour the vacuum character of this subsector (akin to the threshold levels [36]) since we have the same number of fugacities on the unitary and non-unitary side. The numerator of the non-vacuum character of the unitary theory is

$$\chi_{j(0,0)}(q,\omega) - \chi_{j(1,0)}(q,\omega) = q^{1/3} \sum_s \left[ q^{12s^2+4s}(\omega^{3s+1/2} - \omega^{-3s-1/2}) - q^{12s^2+8s+1}(\omega^{3s+1} - \omega^{-3s-1}) \right]. \tag{70}$$

This will be related to $\chi_1^{(1)}(q,\omega)$ due to (68) in the unflavoured limit,

$$\chi_1^{(1)}(q,\omega) = 2q^{1/3} \sum_s \left[ q^{12s^2+4s}(\omega^{6s+1} - \omega^{-6s-1}) - q^{12s^2+8s+1}(\omega^{6s+2} - \omega^{-6s-2}) \right]. \tag{71}$$

If these two flavoured characters are equal upto a function of $\omega$, then the relation can be expressed as

$$\chi_1^{(1)}(q,\omega) = \chi_2^{su(2)}(\omega)\left(\chi_{j(0,0)}(q,\omega) - \chi_{j(1,0)}(q,\omega)\right), \quad \omega = e^{\frac{2\pi i k}{3}}. \tag{72}$$

For this to be true, the ratios,

$$r_1(s,z) = \frac{\sin(2\pi z(6s+1))}{\sin(\pi z(6s+1))} = 2\cos(6\pi zs + \pi z),$$

$$r_2(s,z) = \frac{\sin(2\pi z(6s+2))}{\sin(\pi z(6s+2))} = 2\cos(6\pi zs + 2\pi z), \tag{73}$$

should be equal and independent of $s$. At $z = k/3$ for $k \in \mathbb{Z}_{\geq 0}$ the ratios are independent of $s$, but not always equal. At only $k = 0$ and $k = 2$ are the ratios equal, where the former

---

[21]The common multiplicative scaling factor $u = 4$ for the even characters is reduced to $u = 2$ for this subsector. The modular-$S$ matrix is in a block diagonal form anyways, so the two subsectors can have two different scaling factors.

value yields the relation between the unflavoured characters (68). This proves (72) at discrete fugacities $\omega = e^{\frac{2\pi i k}{3}}$, where $k = 0$ or $k = 2$.

As a final remark, we see that the set of independent even unflavoured characters of this theory decomposes into a couple of sub-sectors. This can be explicitly seen if we write the $S$-matrix for the characters. For convenience, let us rewrite the S-matrix of the $m = -5/4$ theory again,

$$
S = \frac{1}{2\sqrt{2}}\begin{pmatrix} -1 & 1 & 1 & -1 & 1 \\ 1 & -1 & 1 & -1 & -1 \\ 2 & 2 & \sqrt{2} & \sqrt{2} & 0 \\ -2 & -2 & \sqrt{2} & \sqrt{2} & 0 \\ 2 & -2 & 0 & 0 & 2 \end{pmatrix}. \tag{74}
$$

Let us label the characters as $\chi_1 = \chi_{j(0,0)}$, $\chi_2 = \chi_{j(1,0)}$, $\chi_3 = \chi^+_{j(0,1)}$, $\chi_4 = \chi^+_{j(1,1)}$, and $\chi_5 = \chi^+_{j(1,2)}$, which form the basis of the above $S$-matrix. The conformal dimensions of the primary fields corresponding to $\chi_1$ and $\chi_2$ differ by an integer. Although the $S$-matrix (74) is not manifestly block-diagonal, we can utilise the fact that we can take linear combinations of $\chi_1$ and $\chi_2$ to write new characters. A rearrangement of the column vector after taking the linear combination gives

$$
\begin{pmatrix} (\chi_1+\chi_2)(-1/\tau) \\ \chi_3(-1/\tau) \\ \chi_4(-1/\tau) \\ (\chi_1-\chi_2)(-1/\tau) \\ \chi_5(-1/\tau) \end{pmatrix} = \frac{1}{2\sqrt{2}}\begin{pmatrix} 0 & 2 & -2 & 0 & 0 \\ 2 & \sqrt{2} & \sqrt{2} & 0 & 0 \\ -2 & \sqrt{2} & \sqrt{2} & 0 & 0 \\ 0 & 0 & 0 & -2 & 2 \\ 0 & 0 & 0 & 2 & 2 \end{pmatrix}\begin{pmatrix} (\chi_1+\chi_2)(\tau) \\ \chi_3(\tau) \\ \chi_4(\tau) \\ (\chi_1-\chi_2)(\tau) \\ \chi_5(\tau) \end{pmatrix}. \tag{75}
$$

Thus, we have two independent subsectors in the even sector of the fractional level $m = -5/4$. We will explore whether a subsector can be extracted for theories with quasi-character in their even sector in appendix C. We return to the last class of special fractional levels which occur at half-odd integer values in the next subsection.

## 3.3 Positive half-integer levels

In this subsection, we will look at the other infinite series of theories with admissible characters. These theories are parametrised by $(p = 2N+3, u = 2)$ with $N \in \mathbb{Z}_{>0}$. The corresponding levels are half odd integers $m = p/u - 2 = (2N-1)/2$. We have checked that the $q$-expansions of all the characters for $5 \leq p \leq 29$ have positive coefficients (up to $\sim q^{2000}$). As an aside, the $(p = 3, u = 2)$ theory has a quasi-character but we will deal with this case separately. For now, we focus on the unitarisation of the entire $(p = 2N + 1, u = 2)$ sequence with $N \in \mathbb{Z}_{>0}$.

The central charge of these theories is given by

$$
c = 3 - \frac{12}{2N + 1}. \tag{76}
$$

Since we have $u = 2$, $k$ can take two values 0 and 1. The number of independent even characters for $p = 2N + 1$ is $3N$. The $2N$ characters corresponding to $k = 0$ are equal to the half-integer spin characters of $\widehat{su(2)}_{4N}$ theory. The remaining $N$ even characters $\chi^+_{j(n,1)}$, corresponding to $k = 1$ can be written in terms of the difference of integer spin characters of the same affine Lie algebra $\widehat{su(2)}_{4N}$.

The spins $j(n, k)$ are

$$
j(n, 0) = \frac{1}{2}n,
$$

$$
j(n, 1) = \frac{1}{2}(n - N) - \frac{1}{4}, \tag{77}
$$

with the conformal dimensions,

$$h(n,0) = \frac{n(n+2)}{2(2N+1)},$$

$$h(n,1) = \frac{(n-N)^2 + (n-N) - 3/4}{2(2N+1)}. \tag{78}$$

The minimum conformal dimension among them is

$$h_{min} = h(N,1) = \frac{-3}{8(2N+1)}. \tag{79}$$

Following the usual unitarisation procedure, *i.e.*, identifying the character $\chi^+_{j(N,1)}$ with the character of the identity of the unitary theory gives the unitarised central charge

$$c_U = 3 - \frac{12}{2N+1} + \frac{9}{2N+1} = \frac{3(4N)}{4N+2}. \tag{80}$$

This is the central charge of $\widehat{su(2)}_{4N}$ WZW models from the traditional Sugawara construction. The unitarised conformal dimensions are obtained by adding $h_{min}$ to all the conformal dimensions of the original theory. For $k = 0$, set

$$h_U(n,0) = \frac{(n+1/2)((n+1/2)+1)}{4N+2}. \tag{81}$$

We identify the spin $l/2 = n + 1/2$ with odd $l$ values of the representation of $\widehat{su(2)}_{4N}$. The other set of characters (with $k = 1$) have dimensions

$$h_U(n,1) = \frac{(n-N)(n-N+1)}{4N+2}. \tag{82}$$

The identification of spins in this case is $l/2 = (n-N)$ with even $l$. We can choose the set $N \leq n \leq 2N-1$ so that $l$ is positive.

The $k = 0$ unflavoured characters match with the unflavoured characters of integrable highest weight representations of $\widehat{su(2)}_{4N}$ with spins $l/2 = n + 1/2$,

$$\chi_{j(n,0)}(q) = \frac{1}{\eta(q)^3} q^{(2n+2)^2/4(4N+2)} \sum_{s \in \mathbb{Z}} q^{(4N+2)s^2 + s(2n+2)}(2(4N+2)s + 2n + 2)$$
$$= \chi^{(4N)}_{2n+1}(q) = \chi^{(4N)}_l(q). \tag{83}$$

The characters with $k = 1$, on the other hand, can be written as the difference of the characters $(\chi^{(4N)}_l - \chi^{(4N)}_{4N-l})$ of the integrable highest weight representations of the unitarised theory with spins $l = 2(n-N)$ and $4N-l = 6N-2n$ respectively, with $n$ restricted to take values in $N \leq n \leq 2N-1$ for both cases.

We will use the variable $l$ instead of $n$ to write the $q$-expansion of the character,

$$\chi^+_{j(n,1)}(q) = \frac{2q^{(l+1)^2/4(4N+2)}}{\eta^3(q)} \sum_s \Bigg[ q^{s^2(4N+2)+s(l+1)}(s(4N+2) + (l+1)/2)$$
$$- q^{s^2(4N+2)-s(l+1)-s(4N+2)+n+1}\left(s(4N+2) - \frac{l+1+4N+2}{2}\right)\Bigg]$$
$$= \chi^{(4N)}_l(q) - \chi^{(4N)}_{4N-l}(q), \tag{84}$$

where we have used the map $\sigma_0$ takes $l/2 \to -l/2 - 1$ to rearrange the characters with $l$ an even integer. It is not obvious from this expression for the characters that the resulting

character $\chi^+_{j(n,1)}$ has manifestly positive coefficients. In fact, there is a counter-example, at the level $m = -1/2$, i.e. $N = 1$, the character $\chi^+_{j(0,1)} = -\chi^+_{j(1,1)}$ has the $q$-expansion[22]

$$\chi^+_{j(1,1)}(q) = q^{-1/12}\left(1 - 2q^1 + q^2 - 2q^3 + 4q^4 - 4q^5 + 5q^6 - 6q^7 + 9q^8 + \mathcal{O}(q^9)\right). \qquad (85)$$

The unitarised theory has 3 independent even characters with the vacuum character given by eq. (85). We have checked for large powers of $q$, that the quasi-character $\chi^+_{j(1,1)}$ alternates between positive and negative signs which belong to the class D of quasi-characters and the three character theory in class DAA as defined in section 4.3 of [32].[23]

For the theories with only admissible characters, i.e., $p = 2N + 3$ ($N \geq 1$), given the fact that the character $\chi^+_{j(n,1)}$ is obtained by subtraction of two characters, it is not guaranteed that it will have manifestly positive definite coefficients in the $q$-expansion. Nevertheless, except in the case of $m = -1/2$, it is possible to show that the coefficients of $\chi^{(4N)}_l$ are consistently larger than those for $\chi^{(4N)}_{4N-l}$ for $N \geq 2$ and hence the character $\chi^+_{j(n,1)}$ is an admissible character. Although we do not have explicit proof of this assertion, we have checked this up to sufficiently high orders in the $q$-expansion. For example, consider the $N = 2$ case, which has the characters

$$\chi^+_{j(2,1)}(q) = q^{-1/10}\left(1 + 3q + 6q^2 + 3q^3 + 9q^4 + 4q^5 + 18q^6 + 9q^7 + 30q^8 + 12q^9 + \mathcal{O}(q^9)\right),$$
$$\chi^+_{j(3,1)}(q) = q^{1/10}\left(3 + 2q + 6q^2 + 3q^3 + 12q^4 + 6q^5 + 24q^6 + 9q^7 + 39q^8 + 18q^9 + \mathcal{O}(q^9)\right). \qquad (86)$$

For higher $N$ the coefficients increase at faster rates. The subtraction of the characters also means that the character $\chi^{(4N)}_{2N}$ will never appear in the partition function.

Let us continue with the example of $N = 2$. This is an illustrative example. All higher $N$ values work out essentially in an identical manner.

$$\mathcal{Z} = |\chi_{j(0,0)}|^2 + |\chi_{j(1,0)}|^2 + |\chi_{j(2,0)}|^2 + |\chi_{j(3,0)}|^2 + \frac{1}{2}|\chi^+_{j(2,1)}|^2 + \frac{1}{2}|\chi^+_{j(3,1)}|^2. \qquad (87)$$

The factor of $1/2$ in front of a couple of characters is potentially problematic for we need the partition integers to be positive integers. This can be avoided by rescaling the partition function by a factor of 2,

$$\tilde{\mathcal{Z}} = 2\mathcal{Z} = |\chi^+_{j(2,1)}|^2 + |\chi^+_{j(3,1)}|^2 + 2|\chi_{j(0,0)}|^2 + 2|\chi_{j(1,0)}|^2 + 2|\chi_{j(2,0)}|^2 + 2|\chi_{j(3,0)}|^2. \qquad (88)$$

At this point, it appears that we have a couple of choices for the vacuum character, namely, it could be $\chi^+_{j(2,1)}$ or $\chi^+_{j(3,1)}$, but a glance at eq.(86) would convince us that it is the former one that is a vacuum character and not the latter as predicted from eq. (79).

Since the characters are written as subtraction of two integrable highest weight characters, the partition function when expanded in terms of the integrable characters have negative coefficients. The presence of these negative signs does not allow the description of the partition function as an ordinary CFT.

$$\tilde{\mathcal{Z}} = 2\mathcal{Z} = |\chi^{(8)}_0 - \chi^{(8)}_8|^2 + |\chi^{(8)}_2 - \chi^{(8)}_6|^2 + 2|\chi^{(8)}_1|^2 + 2|\chi^{(8)}_3|^2 + 2|\chi^{(8)}_5|^2 + 2|\chi^{(8)}_7|^2. \qquad (89)$$

However, partition functions with negative coefficients are quite common in defect CFT, but the defect partition functions have different modular properties. In this case, we have a partition

---

[22]It is tempting to think of this expression not as a character but as a Witten Index of a theory with broken supersymmetry as the alternating signs are reminiscent of the operator $(-1)^F$ insertion in the character. However, we do not have a reason to believe supersymmetry is at play here.

[23]The $m = -1/2$ case has been studied in detail [11–13, 15–18] and was the second example where the logarithmic nature of $\widehat{su(2)}_{-1/2}$ was concretely established.

function which is modular invariant under $SL(2, \mathbb{Z})$, and as a result, it is not a partition function of a defect CFT. This effectively rules out the possibility of the half-integer level $\widehat{su(2)}$ highest weight states being related to defect CFT.

Although, we find an interesting correspondence between the characters at the fractional level $m = (2N-3)/2$ and the character at the level $M = 4N$ of the $\widehat{su(2)}$ CFT, the choice of the vacuum in the unitary theory poses a new puzzle. In general, for the tentative unitary RCFT with the unflavoured characters as the even unflavoured characters at half-odd integer levels, we find that the RCFT has a unique vacuum but the modular invariant partition function has negative multiplicities when the character is expressed in terms of the $\widehat{su(2)}_{4N}$ characters, and the naive Verlinde fusion coefficients are not well-defined.

To understand this better, let us look at the modular transformation of the characters $\chi_{j(n,1)}^{+}$ in the 'unitary' theory,

$$\chi_{j(n,1)}^{+}(-1/\tau) = 2\left(\sum_{n' \in \Gamma} S_{n1,n'1}^{+} \chi_{j(n',1)}^{+}(\tau) + \sum_{n'} S_{n1,n'0} \chi_{j(n',0)}(\tau)\right), \quad (90)$$

where the modular $S$-matrix element $S_{n1,n'0}$ is given by eq. (37) and

$$S_{n1,n'1}^{+} = \frac{-i(-1)^{n+n'+N}}{\sqrt{2N+1}}\left[\sin\left(\pi(n'+1)\right)\cos\left(\frac{\pi}{2N+1}(n'+1)(2n-2N+1)\right)\right] = 0. \quad (91)$$

Notice, $S_{n1,n'1}^{+}$ vanishes for all $n, n'$, and the vacuum character of the unitarised theory belongs to this sector. It therefore follows that the $S$-matrix elements involving the vacuum character $\chi_{j(N,1)}^{+}$ and any character belonging to the $\chi_{j(n,1)}^{+}$ set vanishes. Since the $S$-matrix is not unitary, we can not straight away impose Verlinde fusion rules. However, using standard methods [24], we can write the unitary $S$-matrix which will also have $S_{N1,N1} = 0$ since the vacuum is unique. In particular, since some matrix elements in the vacuum row of the modular $S$-transformation vanishes, the corresponding fusion coefficients are not well-defined. This leads to the conclusion that the tentative theory will not be an RCFT.

For completeness, we also write the modular $S$-transformation of $\chi_{j(n,1)}^{+}$ where we will relabel $n = \frac{l}{2} + N$ and $n' = \frac{l'-1}{2}$ so that we can relate them to the integrable representations of $\widehat{su(2)}$ at level $4N$,

$$\chi_{j(l/2+N,1)}^{+}(-1/\tau) = \frac{2}{\sqrt{2N+1}} \sum_{l' \in \text{odd}} \sin\left(\frac{\pi}{4N+2}(l+1)(l'+1)\right) \chi_{j(l'/2-1/2,0)}(\tau). \quad (92)$$

Since these characters are written in terms of subtraction of two characters of $\widehat{su(2)}$ at level $M = 4N$ theory, we write the modular S-transformation as

$$\chi_l^{(4N)} - \chi_{4N-l}^{(4N)}(-1/\tau) = \frac{2}{\sqrt{2N+1}} \sum_{l' \in \text{odd}} \left[\sin\left(\frac{\pi}{4N+2}(l'+1)(l+1)\right)\right]\chi_{l'}^{(4N)}(\tau). \quad (93)$$

In summary, we find that the unitary RCFT does not have correct fusion coefficients due to vanishing matrix elements involving the vacuum character. It may be interesting to explore the possibility of expanding the Hilbert space of the CFT in such a way that the modular $S$-matrix has non-vanishing vacuum row and column entries.

There is an alternate way of salvaging these theories by looking for a suitable subsector of the theory which forms a closed set under the fusion algebra and admits a modular invariant partition function comprising only the characters in the subsector. While this seems a worthwhile exercise for the theories containing the quasi-characters, it is easy to see that there is no

such suitable subsector in any theory with $u = 2$. The argument is as follows. The primaries with the characters $\chi_{2l}^{(4N)} - \chi_{4N-2l}^{(4N)}$ have integer spins. One may be tempted to think that the integer spin subsector can very well have closed fusion algebra, but for the $u = 2$ case, we have seen that the modular $S$-transformation eq. (93) relates the integer spin sector to the half odd-integer spin sector. On the other hand, the sector with characters $\chi_{2l+1}^{(4N)}$ have half odd-integer spins, which by themselves do not close under the $\widehat{su(2)}_{4N}$ fusion algebra. A little generalisation of this argument can show that no mixing of half odd-integer and integer spin subsectors can have a closed fusion algebra.

## 4 Conclusions and Discussion

In this paper, we complete the picture of the even characters of the highest weight representations of the $\widehat{su(2)}$ log CFTs at fractional levels in the well-defined unflavoured limit by implementing bootstrap constraints on the even characters. We found admissible character vectors at only three special classes of fractional levels. These special fractional levels are classified into two infinite sequences, the threshold level sequence with $(p = 2, u = 2N + 1)$ and the sequence with positive half-integer levels with $(p = 2N + 3, u = 2)$ where $N \in \mathbb{Z}_{>0}$, and the isolated level $(p = 3, u = 4)$. Our classification of the $\widehat{su(2)}$ admissible levels includes the two new mappings for the isolated level and the half-odd integer levels. The isolated level is the most curious case as its characters with correct fusion rules can be decomposed into two independent modular invariant sectors, one sector corresponds to $\widehat{so(5)}_1$ and the other to $\widehat{su(2)}_1$. The reason for such a decomposition is not clear to us. We have expanded the unflavoured correspondence for the 2-character subsector with $\widehat{su(2)}_1$ algebra to a flavoured correspondence for the non-vacuum character of the unitary theory at two discrete values of fugacity. We find the even characters of the half-odd integer levels to be related to the difference of characters of the $\widehat{su(2)}_{4N+4}$ algebra. Although not manifest, we have checked their positivity to a reasonably large power of $q$-expansion. However, when $\widehat{su(2)}_{4N+4}$ integrable representation characters are used as the building blocks of the partition function, it has negative multiplicities. The negative multiplicities in a modular invariant partition function do not correspond to a physical partition function. Moreover, since the Verlinde fusion rules are ill-defined, the characters do not represent any RCFT as they are incompatible with an MTC structure. For the threshold cases, we have shown the positivity and integrality of the $q$-series coefficients by writing these characters as sums of integrable highest weight characters of the $\widehat{su(2)}_{4N}$ WZW models, reaffirming an old observation [22]. The correspondence between the unflavoured even characters at the threshold levels and the unflavoured characters of $\widehat{su(2)}_{4N}$ was already extended to the flavoured vacuum character of the non-unitary theory to the spin $N$ character of the unitary theory at discrete flavour fugacities in [36]. However, note that the modular data alone is not enough to completely classify the tentative RCFT. In other words, although we have introduced discrete flavour fugacity for the $\widehat{su(2)}$ sectors, we have ignored the flavour fugacity while classifying the RCFT.

From another point of view, our results regarding the even sector of the Kač-Wakimoto characters at the admissible levels of $\widehat{su(2)}$ current algebra classify the fractional levels into two sets: with and without quasi-characters, i.e., in many cases the $q$-expansion is a vector valued modular function with nice modular properties which do not satisfy the positivity of the $q$-series coefficients. Apart from the special fractional levels mentioned above, we find quasi-characters at every other level. Note that the modules of fractional level theories are equivalent to formal power series in $q$ and $\omega$, and within its radius of convergence to the ana-

lytically continued characters of Kač-Wakimoto. In this paper, the difference of the analytically continued characters, i.e, the even characters were expanded in the region $1 < |\omega| < |q|^{-1}$ for $|q| < 1$. Equivalently they could be expanded in the region $|q| < |\omega| < 1$ for $|q| < 1$ due to the linear dependence, for example (29). Both of the expansions have a well-defined $\omega \to 1$ limit with an integral $q$-series. Surely, there are other regions of expansions of the Kač-Wakimoto character where we will not encounter quasi-characters but those characters are not realised by an RCFT. In particular when the expansions are equivalent to the modules of the fractional level theories. The even characters with the naive radius of convergence have been used in earlier works, for example, [33, 35, 36], however, it remains to be seen if we can draw any conclusions between the unitary and non-unitary theories while expanding in other regions of convergence.

The fact that the even sector of only a few special fractional level theories admits admissible unflavoured character vectors points to some intricate relations between the integrable and fractional level $\widehat{su(2)}$ theories. It would also be interesting to explore the consequences of our results on the corresponding 4-dimensional SCFT. In light of [33, 47, 48], we expect non-trivial constraints on which non-unitary $\widehat{su(2)}$ chiral algebras might be relevant, particularly in the presence of non-local operators such as lines or surface defects. We hope to shed some light on the correspondence soon.

We hasten to point out that the existence of quasi-characters is not necessarily a lost cause. As shown earlier for the two [26] and three [32] component character vectors, admissible characters can be obtained from the quasi-characters either by adding two quasi-characters with proper coefficients or by multiplication of an appropriate power of the modular invariant $j$-function. It will be interesting to answer whether the quasi-characters found in the even sector of fractional level theories are related to admissible characters of some RCFT.

# Acknowledgments

We are immensely thankful to Dileep P. Jatkar for valuable suggestions, discussions, and guidance while preparing the first draft. We benefited greatly from illuminating comments and suggestions by Sujay Ashok, Subramanya Hegde, Sunil Mukhi and K.P. Yogendran. We are grateful to the anonymous referees who commented on a previous version and the incredibly constructive reviews from the referee at SciPost, which guided the current version of the paper.

# A   Quasi-characters examples

In this appendix, we list examples of quasi-characters to shed light on the types of quasi-characters we encountered. All the characters have only a finite number of positive or negative coefficients respectively. This implies the sign flips only once in the $q$-expansion.

We encountered only one example of a quasi-character with infinitely many positive or negative signs, for $\widehat{su(2)}_{-1/2}$ as discussed in section 3.

## A.1   $n = 0, k = \lceil u/p \rceil$

We find that we always get a quasi-character for $(n = 0, k = \lceil u/p \rceil)$ labels except at the special admissible levels described in section 3. These characters typically do not contain integer entries but have a common denominator. They all have integer coefficients after they are multiplied by $u$.

Let us now consider a few explicit examples. We will list them using the parameter $u$ for $p = 3, 4$.

**$p = 3$ sequence:**

- $u = 5$, $m = -7/5$, $c = -7$,

$$\chi^+_{(1,3)}(q) = q^{-13/120}(1 + 3q - 2q^2 - 11q^3 - 48q^4 - 134q^5 - 321q^6 - \mathcal{O}(q^7)), \quad \text{(A.1)}$$

- $u = 7$, $m = -11/7$, $c = -11$,

$$\chi^+_{(1,4)}(q) = q^{-13/168}(2 + 6q + 18q^2 + 28q^3 + 54q^4 + 72q^5 - \cdots - 82q^8 + \mathcal{O}(q^9)), \quad \text{(A.2)}$$

- $u = 8$, $m = -13/8$, $c = -13$,

$$\chi^+_{(1,5)}(q) = q^{-11/96}(1 + 3q + 9q^2 + 5q^3 - 45q^4 - 153q^5 - 219q^8 - \mathcal{O}(q^7)). \quad \text{(A.3)}$$

**$p = 4$ sequence:**

- $u = 3$, $m = -2/3$, $c = -3/2$,

$$\chi^+_{(2,2)}(q) = q^{-5/48}(1 - 4q - 12q^2 - 41q^3 - 103q^4 - 249q^5 - 518q^6 - \mathcal{O}(q^7)), \quad \text{(A.4)}$$

- $u = 5$, $m = -6/5$, $c = -9/2$,

$$\chi^+_{(2,3)}(q) = q^{-1/80}(3 + 9q + 14q^2 + 27q^3 + 36q^4 + 38q^5 - 117q^6 - \mathcal{O}(q^7)), \quad \text{(A.5)}$$

- $u = 7$, $m = -10/7$, $c = -15/2$,

$$\chi^+_{(2,5)}(q) = q^{-13/112}(1 + 3q - 6q^2 - 23q^3 - 84q^4 - 222q^5 - 544q^6 - \mathcal{O}(q^7)). \quad \text{(A.6)}$$

For additional examples, we refer the reader to the table D where we have provided data for $p = 3, 4, \cdots, 10$. In most of these examples, we see that the sign flip occurs at fairly small powers in the *q-expansion*. However, we have checked all these cases up to $\sim q^{100}$ to see if there is another sign flip. We generically find that there is no instance except the $(p = 3, u = 2)$ case corresponding to $m = -1/2$ mentioned above. In the table D among other things, we have listed the first coefficient of the $q$-series including its sign as well as the sign of the $90th$ coefficient of the $q$-series to indicate that the sign flip in all those cases up to that order happens only once. As can be seen from these two data points that there is a single sign flip for all cases except the $(p = 3, u = 2)$ case.

## A.2 Multiple quasi-characters

As mentioned earlier, most of the fractional level cases except the special fractional levels mentioned in sec. 3 possess at least one quasi-character. But, as can be seen from the table D that there are several cases where we have multiple quasi-characters. We will look at some examples of this type here.

For the level $m = -10/7$, we encounter a second quasi-character with $b_+ = -5$, which takes us beyond the criteria of finding a quasi-character at $n = 0$, $k = \lceil u/p \rceil$.

$$\chi^+_{(2,4)}(q) = q^{-11/112}(5 + 15q + 45q^2 + 91q^3 + 198q^4 + \cdots - 51908q^{21} + \mathcal{O}(q^{22})). \quad \text{(A.7)}$$

In this example we encounter two quasi-characters in the character vector. As we progress to larger values of $u$ and $p$ we encounter more quasi-characters for the set of characters.

- $p = 6$, $u = 5$, $m = -4/5$, $c = -2$.

  The first one is the quasi-character satisfies $n = 0$ and $k = \lceil u/p \rceil$,

  $$\chi^+_{(4,4)}(q) = q^{-7/60}(1 - 8q - 24q^2 - 77q^3 - 191q^4 - 453q^5 - 967q^6 - \mathcal{O}(q^7)). \quad \text{(A.8)}$$

  The other two quasi-characters are,

  $$\chi^+_{(4,3)}(q) = q^{17/60}(7 + 21q + 46q^2 + 103q^3 + 204q^4 + \cdots - 30886q^{20} - \mathcal{O}(q^7)), \quad \text{(A.9)}$$

  $$\chi^+_{(3,3)}(q) = q^{-11/120}(2 + 6q + 18q^2 + 44q^3 + 80q^4 + \cdots - 240q^{11} - \mathcal{O}(q^7)). \quad \text{(A.10)}$$

In the supplementary table D, we list about the first 20 characters independent up to the relation (35) found for each value of $p$ using the $q$-expansion 41 up to $q^{100}$. In the table, $a_0$ is the leading order coefficient of the character, and $a_s$ is the term when the sign flips, i.e., the character $\chi(q) = q^{h-c/24}(-a_0 - a_1q - \cdots - a_{s-1}q^{s-1} + a_sq^s + \cdots)$.

# B  Asymptotic behaviour of $q$-series

To derive the conditions for the positivity of coefficients, we arrange the $q$-expansion in such a way that different $q$-series have different leading behaviour and have no overlapping exponents. We will write the expansion for the characters with $k \neq 0$ only since the characters with $k = 0$ are well defined as discussed in section 2.[24]

$$
\begin{aligned}
\chi_{j(n,k)}(\tau) = \frac{b_+}{2u} q^{b_+^2/4a - 1/8} \Bigg[ &\sum_{l=0}^{k(n+1)-1} c_l q^l + \sum_{l=0}^{a+b_--1} \left( c_{k(n+1)+l} - \frac{b_-}{b_+} c_l \right) q^{k(n+1)+l} \\
&+ \sum_{l=0}^{b_+-k(n+1)-b_--1} \left( c_{k(n+1)+a+b_-+l} - \frac{b_-}{b_+} c_{a+b_-+l} - \left( \frac{2up}{b_+} + \frac{b_-}{b_+} \right) c_l \right) q^{k(n+1)+a+b_-+l} \\
&+ \sum_{l=0}^{-2b_+-1} \left( c_{a+b_++l} - \frac{b_-}{b_+} c_{a+b_+-k(n+1)+l} - \left( \frac{2up}{b_+} + \frac{b_-}{b_+} \right) c_{b_+-k(n+1)-b_-+l} \right. \\
&\left. + \left( \frac{2up}{b_+} + 1 \right) c_l \right) q^{a+b_++l} \\
&+ \sum_{l=0}^{-b_-+k(n+1)+b_+-1} \left( c_{a-b_++l} - \frac{b_-}{b_+} c_{a-b_+-k(n+1)+l} - \left( \frac{2up}{b_+} + \frac{b_-}{b_+} \right) c_{-b_+-b_--k(n+1)+l} \right. \\
&\left. + \left( \frac{2up}{b_+} + 1 \right) c_{-2b_++l} + \left( -\frac{2up}{b_+} + 1 \right) c_l \right) q^{a-b_++l} \\
&+ \sum_{l=0}^{(2+1)(a+b_-)-1} \left( c_{k(n+1)+a-b_-+l} - \frac{b_-}{b_+} c_{a-b_-+l} - \left( \frac{2up}{b_+} + \frac{b_-}{b_+} \right) c_{-2b_-+l} \right. \\
&\left. + \left( \frac{2up}{b_+} + 1 \right) c_{k(n+1)-sb_--b_++l} + \left( \frac{-2up}{b_+} + 1 \right) c_{k(n+1)-b_-+b_++l} \right. \\
&\left. + \left( \frac{2up}{b_+} - \frac{b_-}{b_+} \right) c_l \right) q^{k(n+1)+a-b_-+l} + \Sigma \Bigg]. \quad \text{(B.1)}
\end{aligned}
$$

---

[24]We will use the choice $b_+ < 0$.

The $c_l$ is the coefficient of the series $1 + 3q + 9q^2 + \cdots = \sum_i c_i q^i$ obtained from expanding the denominator of the character as a $q$-series on the numerator. Finally, $\Sigma$ is the correction to the character at order $q^{4a+2b_-+k(n+1)}$, *i.e.*, due to $s = 2$ terms in the sum (41). Typically, eq. (B.1) is a good approximation even at higher orders if we let the sum over $l$ run over larger values in the last summation.

## C   Closed subsets without quasi-characters?

In this section, we report our preliminary investigations into the problem of finding sub-sectors which are devoid of quasi-characters. So far we have been looking at theories at those fractional levels, which only have admissible characters. However, as we have seen, there are theories at other fractional levels which possess at least one quasi-character. It is easy to see that these theories have a modular invariant partition function with the inclusion of the quasi-character, but the presence of the quasi-character in the modular invariant makes difficult an interpretation of the partition function in terms of state counting of a physical system.

To remedy this problem, we try to find subsectors of these theories. The subsectors are an admissible subset of characters closed under the fusion algebra equipped with a modular invariant partition function. It is a priori not guaranteed that theories with such sub-sectors without quasi-characters exist at all. The reason for this suspicion stems from the observation that the modular $S$-matrix for these theories typically does not decompose into a manifestly block-diagonal form so that quasi-characters can be isolated. It is straightforward to show that if the $S$-transformation matrix is in a block-diagonal form, it is a sufficient and necessary condition for the existence of a closed sub-sector.

We have seen a glimpse of a couple of sub-sectors when we studied the case of $m = -5/4$. Note that the linear combination of the characters provided a new basis, in terms of which we could block-diagonalise the $S$-matrix. In general a block-diagonal $S$-matrix is not easy to achieve. It is even more difficult when the eigenvalues of the modular $T$-transformation are non-degenerate.

In the remaining part of this section we set up two simple criteria. The admissible fractional level theories with quasi-characters which satisfy the criteria do not have a subsector without the quasi-character.

### C.1   First criterion

One simple way a quasi-character with labels $(n_q, k_q)$ would decouple from a subsector is when $S_{(n_q k_q),(nk)} = 0$, $\forall (n, k) \neq (n_q, k_q)$.[25] This implies that the quasi-character forms a meromorphic (single-character) CFT imposing strict bootstrap constraints on the conformal dimension of the quasi-character and the central charge of the theory. Namely, $c - 24h_q = 0 \bmod(8)$. We performed a simple check for the quasi-character with labels $(0, \lceil u/p \rceil)$ for $(3 \leq p \leq 30, 3 \leq u \leq 30)$ and did not find any quasi-character of this type which forms a single character CFT. We will therefore consider a weaker criterion of imposing the vanishing of the $S$-matrix element condition only on the sub-sector labelled by $(n, k)$ within all allowed values of $(n, k)$. This will imply that the quasi-character can have non-vanishing $S$-matrix entries with the characters outside the chosen sub-sector.

On the other extreme, if there is no entry of the modular $S$-matrix which vanishes and all characters have non-degenerate eigenvalues of the modular $T$-transformation then it is guaranteed that the $S$-matrix can not be reduced to a block-diagonal form. This implies that a

---

[25]Here $S_{(n_q k_q),(nk)}$ denotes the modular $S$-matrix for the entire even sector and not just $k = 0$ sector.

subsector is not possible in such a theory. This is the first criterion we propose in this subsection.

To explore whether we can find such theories with no subsectors, we need to find the level $(p, u)$ which do not have any vanishing entry of the modular $S$-matrix for the even sector. Since the even characters have two classes, $k = 0$ and $k \neq 0$, we have two conditions corresponding to the $S$-matrix elements for the two classes. We will evaluate the constraints separately as they are independent.

We first set

$$S_{nk,n'k'} = 0, \tag{C.1}$$

to find out what constraints this condition imposes on $(n, k)$, $(n', k')$ and $(p, u)$.[26] The constraint coming from eq. (C.1) is

$$\frac{u(n+1)(n'+1)}{p} = r', \quad r' \in \mathbb{Z}^+,$$
$$(n+1)(n'+1) = rp, \quad r \in \mathbb{Z}^+. \tag{C.2}$$

The second equality follows from the first by using the fact that $p$ and $u$ are co-prime. This constraint is independent of $k$ and $k'$. We can therefore derive conditions on $n$ and $n'$ irrespective of the value of $k, k'$.

Our first criterion comes from the case when $p$ is a prime number. Since $p$ cannot be factored, eq. (C.2) implies $p$ is either $n+1$ or $n'+1$. However, values of $n$ and $n'$ are bounded, $0 \leq n, n' \leq p - 2$. Therefore, eq. (C.2) is not satisfied for any prime number $p$. This criterion will be sharpened momentarily when we consider the second condition on the $S$-matrix, but before we do that, let us mention that if $p$ is not a prime number then it is possible to obtain solutions to eq. (C.2).

The non-prime $p$ can be factorised in at least one way as $p = p_1 p_2$, where $p_1$ and $p_2$ are (non-unique) integer factors of $p$ at this step. As can be easily seen, the equation eq. (C.2) can have multiple solutions in general. We easily see that if we choose $p_1$ and $p_2$ such that $p = p_1 p_2$ and $(p_1 + p_2)$ is minimised, we have $\max(r) = (p - p_1)(p - p_2)/p$. It can also be easily seen that for every value of $1 \leq r \leq \max(r)$ there exists at least one solution to eq. (C.2).

Another place where zeros of $S$-matrix can occur comes from the condition $S_{nk,n'k'} = S_{\bar{n}\bar{k},n'k'}$.

This implies a condition on $k$ and $k'$ and this condition is independent of the values $n$ and $n'$,

$$2kk' = (2s - 1)u, \quad s \in \mathbb{N}. \tag{C.3}$$

Since the left-hand side of eq. (C.3) is an even integer, it implies that unless $u$ is an even integer, there are no solutions $k, k'$ of this condition. We then reach an interesting conclusion, combining the results for this eq. (C.3) and the first condition eq. (C.2), that the S-matrix has no zeros for the level parametrised by ($p \in$ prime, $u \in$ odd) which translates to the statement that there are no sub-sectors for these levels, up to the existence of characters with degenerate eigenvalues of the modular $T$-transformation, which by suitable addition of rows of the $S$-matrix could, in principle make a sub-sector possible.[27] We tabulate a few special cases where the dimensions of the characters belonging to the same theory are separated by an integer in table 3. A closer look at the table makes it clear that the criteria works well only for ($p \in$ prime, $u \in$ prime). Since there is no sub-sector, more so without a quasi-character, the

---

[26]For a simpler criterion, we can only look at the ($n_q = 0, k_q = \lceil u/p \rceil$) row of the $S$-matrix, but since there can be more quasi-characters, we will look at a more general constraint.

[27]The characters which transform in the same way under the modular $T$-transform should also transform appropriately under the modular $S$-transformations. Thus, an arbitrary linear combination of the characters is not possible.

Table 3: Theories with ($p \in 3 \leq$ prime $\leq 29, u \in 3 \leq$ odd $\leq 29$) which have characters of degenerate eigenvalue of the modular $T$-transformation.

| p | u |
|---|---|
| 3 | 25 |
| 5 | 9,21,27 |
| 7 | 9,15,25,27 |
| 11 | 9, 15, 21, 25, 27 |
| 13 | 9, 15, 21, 25, 27 |
| 17 | 9, 15, 21, 25, 27 |
| 19 | 9, 15, 21, 25, 27 |
| 23 | 9,15,21, 25, 27 |
| 29 | 9,15, 21,25, 27 |

theories with ($p \in$ prime, $u \in$ prime) can be eliminated. This is the refined version of the first criterion.

We can continue to find solutions to the condition eq. (C.3) for $u = 2w, w \in \mathbb{Z}^+$. This condition can be subdivided into whether $w$ is prime or not. If $w$ is prime, eq. (C.3) can only be satisfied when either $k = w$ or $k' = w$ as $k, k' \leq u/2 = w$. For $k = w$, we have $\lceil w/2 \rceil = (w+1)/2$ number of $k'$ as solutions of this condition. Whereas if $w \notin$ prime we can write $w = w_1 w_2$ as a (non-unique) factorisation of $w$ where $w_1$ and $w_2$ are integer factors of $w$ at this step. Again, we have the set of solutions due to $k = w$, with $s$ up to the same value max($s$), given by max($s$) $= \lceil w/2 \rceil$. However, we will have additional solutions, as compared with $w \in$ prime, due to different factorisations of $w$. For the quasi-character labelled by ($n = 0, k = \lceil u/p \rceil$) we see that the first condition eq. (C.2) does not give any solutions due to the bound $0 \leq n' \leq p - 2$. To locate a subsector with labels ($n', k'$), which decouples the quasi-character, they should satisfy

$$2\lceil u/p \rceil k_s' = (2s - 1)u, s \in \mathbb{N}, \quad \forall\, n'. \tag{C.4}$$

Thus, the solutions to the above condition, in addition to the conformal data, which may allow for suitable linear combinations of the characters and cause the vanishing of some entries of the modular $S$-matrix, should be enough to provide a necessary condition for the existence of a sub-sector. We hope to provide more details elsewhere. Let us look at an example where both parameters $p$, and $u$ are prime numbers, e.g., ($p = 3, u = 5$). This theory satisfies the criterion. This theory has 6 characters in the even sector corresponding to the set $\Gamma$ tabulated below. The $S$-transformation matrix is given below in the ordered basis of the table 4. Note

Table 4: Conformal data for the even sector of ($p = 3, u = 5$). The shaded entry corresponds to the only quasi-character in the even sector.

| $(n, k)$ | $j(n, k)$ | $h(n, k)$ | $h_U(n, k)$ |
|---|---|---|---|
| (0,0) | 0 | 0 | 2/5 |
| (1,0) | 1/2 | 5/4 | 33/20 |
| (0,1) | -3/10 | -7/20 | 1/20 |
| (1,1) | 1/5 | 2/5 | 4/5 |
| (0,2) | -3/5 | -2/5 | 0 |
| (1,2) | -1/10 | -3/20 | 1/4 |

that all entries of the S-transformation are non-zero and the conformal dimensions are not integer separated to allow any simplification.

$$S = \frac{1}{\sqrt{10}} \begin{pmatrix} -1 & -1 & 1 & 1 & -1 & -1 \\ -1 & 1 & -1 & 1 & -1 & 1 \\ 2 & -2 & 2\cos(2\pi/5) & -2\cos(2\pi/5) & -2\cos(\pi/5) & 2\cos(\pi/5) \\ 2 & 2 & -2\cos(2\pi/5) & -2\cos(2\pi/5) & -2\cos(\pi/5) & -2\cos(\pi/5) \\ -2 & -2 & -2\cos(\pi/5) & -2\cos(\pi/5) & -2\cos(2\pi/5) & -2\cos(2\pi/5) \\ -2 & 2 & 2\cos(\pi/5) & -2\cos(\pi/5) & -2\cos(2\pi/5) & 2\cos(2\pi/5) \end{pmatrix}.$$

(C.5)

This $S$-matrix satisfies $S^\dagger M S = M$ where $M = \text{diag}(2, 2, 1, 1, 1, 1)$. We want to identify a possible sub-sector without the quasi-character, but a diagonal modular invariant for a possible subsector theory does not exist as the S-matrix can not be written as a block-diagonal matrix, nor does it allow an Awata-Yamada fusion subalgebra.[28]

## C.2 Second criterion

The ($p = 3, u = 5$) example also illustrates a simple feature. For the original non-unitary theory, the vacuum is degenerate with the degeneracy $u = 5$ and a multiplicity in the number of primaries equal to 2. Since the lowest dimension primary has $|b_+(0, 2)| = 1$, it can be identified with the vacuum because it is non-degenerate. In this example, the lowest dimension primary also happens to be a quasi-character of the type ($n = 0, k = \lceil 5/3 \rceil$). Since the vacuum is the identity element of the fusion algebra, the simple identification of the quasi-character with the vacuum character of the unitary theory prohibits any closed fusion subalgebra for the theory without the quasi-character, provided we do not change the vacuum identification. A natural option is to look for any other vacuum candidate for a sub-sector. But all other characters are degenerate once we have scaled the characters by $u$ to guarantee integrality and preserving the form of the modular $S$-transformation. As we had seen for the ($p = 3, u = 4$) example, once we have identified a subsector, we could also rescale the characters of the subsector without affecting the form of the modular $S$-transformations. Thus, the candidate vacuum must preserve integrality with leading coefficient unity after rescaling by the common factor and be a part of a sub-sector without the quasi-character. In general, this criterion is hard to satisfy for the subsector, although not outright impossible.

Before we state the criterion for a generic level let us consider another example, that of ($p = 4, u = 3$). The ($p = 4, u = 3$) theory has in total 9 primaries and 6 independent even characters. The shaded entry in the table 5 below belongs to ($n, k$) = $(0, 1)$ which is a quasi-character and happens to be the lowest dimension primary. From the conformal data it looks like it has the same fate as that of ($p = 3, u = 5$), but Awata-Yamada (AY) fusion rules allows a closed sub-sector with the fusion algebra,

$$(0, 0) \times (2, 0) = (2, 0),$$
$$(2, 0) \times (2, 0) = (0, 0).$$

(C.6)

These primaries have spins 0 and 1 respectively, hence from the $su(2)$ point of view, their closure under fusion is expected. For this sub-sector, the degeneracy of the vacuum character from the $q$-expansion can be factored out by a rescaling. However, their conformal data does not agree with the general two primary criteria [4]. We also find that the corresponding $2 \times 2$

---

[28]The Awata-Yamada fusion algebra [8] gives the fusion coefficients either 0 or 1 for all fractional level $\widehat{su(2)}$ theories.

Table 5: Conformal data of $(p = 4, u = 3)$. The shaded entry corresponds to the only quasi-character in the even sector.

| $(n,k)$ | $j(n,k)$ | $h(n,k)$ | $h_U(n,k)$ |
|---------|----------|----------|------------|
| (0,0) | 0 | 0 | 1/6 |
| (1,0) | 1/2 | 9/16 | 35/48 |
| (2,0) | 1 | 3/2 | 5/3 |
| (0,1) | -2/3 | -1/6 | 0 |
| (1,1) | -1/6 | -5/48 | 1/16 |
| (2,1) | 1/3 | 1/3 | 1/2 |
| (0,2) | -4/3 | 1/3 | 1/2 |
| (1,2) | -5/6 | -5/48 | 1/16 |
| (2,2) | -1/3 | -1/6 | 0 |

$S$-transformation matrix is singular and the unflavoured partition function composed of the two elements does not exist.[29]

It seems like at an arbitrary level, if the primary with the lowest conformal dimension has a quasi-character, then the corresponding unitarised theory has no closed fusion subalgebra. As discussed above, the natural way out is to identify some other primary (not the lowest dimension primary) as the vacuum of a subsector theory which excludes the quasi-character. Note that $|b_+| = 1$ occurs only once each [22] in the sets $(n, k = 0)$ and $(n, k \neq 0) \in \Gamma$ for the lowest dimension primary. Therefore, there are no other modules with non-degenerate grade zero subspace to begin with, unless there is a common factor in the $q$-expansion of a set of characters, forming a subsector, which can be factored out as we have seen for the example $(p = 3, u = 4)$. A closer look at the $q$-expansion given in (43) or (44) implies that the factor $b_+$ can be factored out while preserving the integrality of the candidate vacuum character if the following two conditions are simultaneously satisfied,

$$\frac{b_-}{b_+} \in \mathbb{Z}, \quad \text{and,} \quad \frac{2up}{b_+} \in \mathbb{Z}. \tag{C.7}$$

Apart from the known solutions with $|b_+| = 1$, an easy solution is the character with the labels $(n = 1, k = 0)$ with $b_+ = -b_- = 2u$, which is a candidate vacuum for a subsector theory. Another solution is the character with the common factor $|b_+| = 2$ and $|b_-| = 2r$ with $r \in \mathbb{N}$, among others. If we can locate a vacuum character from the above criteria, it should have a degeneracy which can be factored out from all the characters in the subsector, that is, it should have the lowest factor which is a divisor of all other degeneracies such that its leading coefficient is unity, preserving integrality of every character in the subsector and the form of modular $S$-transformation.

A great choice to fulfil these criteria is the $(n, k = 0)$ sector as it looks plausible from the point of view of $su(2)$ representation theory (and Awata-Yamada fusion algebra). This subsector has degeneracy $u$ which can be factored out to identify a vacuum and although it does not include a quasi-character, its $S$-matrix elements with the quasi-character given by labels $(0, \lceil u/p \rceil)$ are non-vanishing (C.2). We have expectedly not found an evidence of the existence of a $k = 0$ subsector.[30] Hence if a subsector without the quasi-character exists, it

---

[29]Although there exists another AY sub-sector which excludes only the $(1, k)$ primaries, this subsector is irrelevant because the subsector includes the quasi-character.

[30]Note that the sectors $(n, k = 0)$ and $(n, k \neq 0)$ generally have different degeneracies and we do not expect a mixing. Again, the theories with degenerate eigenvalues of modular $T$-transformation have more options, but we restrict ourselves to the non-degenerate cases.

must be in the $(n, k \neq 0)$ subsector which must include a candidate vacuum.

We have explored a few possibilities of salvaging theories where the minimum of the conformal dimensions is equal to the conformal dimension of the primary associated with the quasi-character. We now propose that the characters associated with these theories do not allow an RCFT description. This proposal rules out a large number of theories. A subset of theories which can be ruled out in this fashion can be located by equating the value of $(n, k)$ for which the conformal dimension is the minimum, with the labels of the quasi-character $(n = 0, k = \lceil \frac{u}{p} \rceil)$. We have found the set of levels

$$(p, u) = (p, Np - 1), \qquad N \in \mathbb{N}, \quad p \geq 3, \tag{C.8}$$

which satisfies the proposed condition. It is worth reiterating that this criterion is based on the assumption that a subsector excluding the quasi-character, which is the vacuum of the unitarised theory, and including a candidate vacuum is in general not possible and the eigenvalues of $T$-transformation are non-degenerate. We hope to provide a refined criterion and more details elsewhere.

## D Supplementary table

Table 6: Quasi-characters.

| S. No. | $p$ | $u$ | $m$ | $(n, k)$ | $a_0$ | $a_s$ | $\text{sign}(a_{90})$ |
|---|---|---|---|---|---|---|---|
| 1 | 3 | 2 | $-1/2$ | $(0, 1)$ | $-1$ | $a_1 = 2$ | $-1$ |
| 2 | 3 | 5 | $-7/5$ | $(0, 2)$ | $-1$ | $a_2 = 2$ | $+1$ |
| 3 | 3 | 7 | $-11/7$ | $(0, 3)$ | $-2$ | $a_8 = 82$ | $+1$ |
| 4 | 3 | 8 | $-13/8$ | $(0, 3)$ | $-1$ | $a_5 = 45$ | $+1$ |
| 5 | 3 | 10 | $-17/10$ | $(0, 4)$ | $-2$ | $a_{11} = 240$ | $+1$ |
| 6 | 3 | 11 | $-19/11$ | $(0, 4)$ | $-1$ | $a_{11} = 77$ | $+1$ |
| 7 | 3 | 11 | $-19/11$ | $(0, 5)$ | $-4$ | $a_{34} = 2003808$ | $+1$ |
| 8 | 3 | 13 | $-23/13$ | $(0, 5)$ | $-2$ | $a_{15} = 2758$ | $+1$ |
| 9 | 3 | 13 | $-23/13$ | $(0, 6)$ | $-5$ | $a_{56} = 940473641$ | $+1$ |
| 10 | 3 | 14 | $-25/14$ | $(0, 5)$ | $-1$ | $a_{10} = 492$ | $+1$ |
| 11 | 3 | 14 | $-25/14$ | $(0, 6)$ | $-4$ | $a_{37} = 5016840$ | $+1$ |
| 12 | 3 | 16 | $-29/16$ | $(0, 6)$ | $-2$ | $a_{19} = 5286$ | $+1$ |
| 13 | 3 | 16 | $-29/16$ | $(0, 7)$ | $-5$ | $a_{57} = 981308226$ | $+1$ |
| 14 | 3 | 17 | $-31/17$ | $(0, 6)$ | $-1$ | $a_{13} = 1806$ | $+1$ |
| 15 | 3 | 17 | $-31/17$ | $(0, 7)$ | $-4$ | $a_{43} = 63653866$ | $+1$ |
| 16 | 3 | 19 | $-35/19$ | $(0, 7)$ | $-2$ | $a_{24} = 110282$ | $+1$ |
| 17 | 3 | 19 | $-35/19$ | $(0, 8)$ | $-5$ | $a_{63} = 6440055585$ | $+1$ |
| 18 | 3 | 20 | $-37/20$ | $(0, 7)$ | $-1$ | $a_{16} = 3417$ | $+1$ |
| 19 | 3 | 20 | $-37/20$ | $(0, 8)$ | $-4$ | $a_{50} = 676392192$ | $+1$ |
| 20 | 3 | 22 | $-41/22$ | $(0, 8)$ | $-2$ | $a_{29} = 570762$ | $+1$ |
| 21 | 3 | 22 | $-41/22$ | $(0, 9)$ | $-5$ | $a_{71} = 116158108500$ | $+1$ |
| 1 | 4 | 3 | $-2/3$ | $(0, 1)$ | $-1$ | $a_1 = 4$ | $+1$ |

Table 6: Quasi-characters.

| S. No. | $p$ | $u$ | $m$ | $(n,k)$ | $a_0$ | $a_s$ | $\text{sign}(a_{90})$ |
|---|---|---|---|---|---|---|---|
| 2 | 4 | 5 | $-6/5$ | $(0,2)$ | $-3$ | $a_7 = 117$ | $+1$ |
| 3 | 4 | 7 | $-10/7$ | $(0,2)$ | $-1$ | $a_2 = 6$ | $+1$ |
| 4 | 4 | 7 | $-10/7$ | $(0,3)$ | $-5$ | $a_{21} = 51908$ | $+1$ |
| 5 | 4 | 9 | $-14/9$ | $(0,3)$ | $-3$ | $a_9 = 204$ | $+1$ |
| 6 | 4 | 9 | $-14/9$ | $(0,4)$ | $-7$ | $a_{45} = 27281095$ | $+1$ |
| 7 | 4 | 11 | $-18/11$ | $(0,3)$ | $-1$ | $a_3 = 1$ | $+1$ |
| 8 | 4 | 11 | $-18/11$ | $(0,4)$ | $-5$ | $a_{23} = 74916$ | $+1$ |
| 9 | 4 | 11 | $-18/11$ | $(0,5)$ | $-9$ | $a_{82} = 919103509269$ | $+1$ |
| 10 | 4 | 13 | $-22/13$ | $(0,4)$ | $-3$ | $a_{13} = 3264$ | $+1$ |
| 11 | 4 | 13 | $-22/13$ | $(0,5)$ | $-7$ | $a_{45} = 190474799$ | $+1$ |
| 12 | 4 | 15 | $-26/15$ | $(0,4)$ | $-1$ | $a_6 = 58$ | $+1$ |
| 13 | 4 | 15 | $-26/15$ | $(0,5)$ | $-5$ | $a_{28} = 856005$ | $+1$ |
| 14 | 4 | 15 | $-26/15$ | $(0,6)$ | $-9$ | $a_{75} = 453611549547$ | $+1$ |
| 15 | 4 | 17 | $-30/17$ | $(0,5)$ | $-3$ | $a_{17} = 18470$ | $+1$ |
| 16 | 4 | 17 | $-30/17$ | $(0,6)$ | $-7$ | $a_{50} = 371458008$ | $+1$ |
| 17 | 4 | 19 | $-34/19$ | $(0,5)$ | $-1$ | $a_8 = 48$ | $+1$ |
| 18 | 4 | 19 | $-34/19$ | $(0,6)$ | $-5$ | $a_{34} = 10392333$ | $+1$ |
| 19 | 4 | 19 | $-34/19$ | $(0,7)$ | $-9$ | $a_{80} = 1111163267691$ | $+1$ |
| 20 | 4 | 21 | $-38/21$ | $(0,6)$ | $-3$ | $a_{21} = 37557$ | $+1$ |
| 21 | 4 | 21 | $-38/21$ | $(0,7)$ | $-7$ | $a_{57} = 3315025959$ | $+1$ |
| 1 | 5 | 3 | $-1/3$ | $(0,1)$ | $-2$ | $a_1 = 2$ | $+1$ |
| 2 | 5 | 4 | $-3/4$ | $(0,1)$ | $-1$ | $a_1 = 6$ | $+1$ |
| 3 | 5 | 6 | $-7/6$ | $(0,2)$ | $-4$ | $a_7 = 12$ | $+1$ |
| 4 | 5 | 7 | $-9/7$ | $(0,2)$ | $-3$ | $a_5 = 50$ | $+1$ |
| 5 | 5 | 7 | $-9/7$ | $(0,3)$ | $-8$ | $a_{38} = 14688216$ | $+1$ |
| 6 | 5 | 7 | $-9/7$ | $(1,3)$ | $-1$ | $a_{14} = 1278$ | $+1$ |
| 7 | 5 | 8 | $-11/8$ | $(0,2)$ | $-2$ | $a_3 = 10$ | $+1$ |
| 8 | 5 | 8 | $-11/8$ | $(0,3)$ | $-7$ | $a_{26} = 395427$ | $+1$ |
| 9 | 5 | 9 | $-13/9$ | $(0,2)$ | $-1$ | $a_2 = 10$ | $+1$ |
| 10 | 5 | 9 | $-13/9$ | $(0,3)$ | $-6$ | $a_{19} = 41868$ | $+1$ |
| 11 | 5 | 9 | $-13/9$ | $(0,4)$ | $-11$ | $a_{81} = 753671188875$ | $+1$ |
| 12 | 5 | 9 | $-13/9$ | $(1,4)$ | $-2$ | $a_{35} = 5057358$ | $+1$ |
| 13 | 5 | 11 | $-17/11$ | $(0,3)$ | $-4$ | $a_{10} = 594$ | $+1$ |
| 14 | 5 | 11 | $-17/11$ | $(0,4)$ | $-9$ | $a_{44} = 17115861$ | $+1$ |
| 15 | 5 | 11 | $-17/11$ | $(1,5)$ | $-3$ | $a_{67} = 12235696960$ | $+1$ |
| 16 | 5 | 12 | $-19/12$ | $(0,3)$ | $-3$ | $a_7 = 90$ | $+1$ |
| 17 | 5 | 12 | $-19/12$ | $(0,4)$ | $-8$ | $a_{35} = 8460776$ | $+1$ |
| 18 | 5 | 12 | $-19/12$ | $(1,5)$ | $-1$ | $a_{31} = 1386570$ | $+1$ |
| 19 | 5 | 13 | $-21/13$ | $(0,3)$ | $-2$ | $a_5 = 36$ | $+1$ |

Table 6: Quasi-characters.

| S. No. | $p$ | $u$ | $m$ | $(n, k)$ | $a_0$ | $a_s$ | sign$(a_{90})$ |
|---|---|---|---|---|---|---|---|
| 20 | 5 | 13 | $-21/13$ | $(0,4)$ | $-7$ | $a_{28} = 1381824$ | $+1$ |
| 21 | 5 | 13 | $-21/13$ | $(0,5)$ | $-12$ | $a_{83} = 1008775829062$ | $+1$ |
| 1 | 6 | 5 | $-4/5$ | $(0,1)$ | $-1$ | $a_1 = 8$ | $+1$ |
| 2 | 6 | 5 | $-4/5$ | $(0,2)$ | $-7$ | $a_{20} = 30886$ | $+1$ |
| 3 | 6 | 5 | $-4/5$ | $(1,2)$ | $-2$ | $a_{11} = 240$ | $+1$ |
| 4 | 6 | 7 | $-8/7$ | $(0,2)$ | $-5$ | $a_8 = 149$ | $+1$ |
| 5 | 6 | 7 | $-8/7$ | $(0,3)$ | $-11$ | $a_{59} = 1640260669$ | $+1$ |
| 6 | 6 | 7 | $-8/7$ | $(1,3)$ | $-4$ | $a_{37} = 5016840$ | $+1$ |
| 7 | 6 | 11 | $-16/11$ | $(0,2)$ | $-1$ | $a_2 = 14$ | $+1$ |
| 8 | 6 | 11 | $-16/11$ | $(0,3)$ | $-7$ | $a_{18} = 32466$ | $+1$ |
| 9 | 6 | 11 | $-16/11$ | $(0,4)$ | $-13$ | $a_{71} = 121595449667$ | $+1$ |
| 10 | 6 | 11 | $-16/11$ | $(1,4)$ | $-2$ | $a_{29} = 570762$ | $+1$ |
| 11 | 6 | 13 | $-20/13$ | $(0,3)$ | $-5$ | $a_{10} = 99$ | $+1$ |
| 12 | 6 | 13 | $-20/13$ | $(0,4)$ | $-11$ | $a_{46} = 106991913$ | $+1$ |
| 13 | 6 | 13 | $-20/13$ | $(1,5)$ | $-4$ | $a_{65} = 15300126568$ | $+1$ |
| 14 | 6 | 17 | $-28/17$ | $(0,3)$ | $-1$ | $a_3 = 13$ | $+1$ |
| 15 | 6 | 17 | $-28/17$ | $(0,4)$ | $-7$ | $a_{21} = 77790$ | $+1$ |
| 16 | 6 | 17 | $-28/17$ | $(0,5)$ | $-13$ | $a_{66} = 42209473672$ | $+1$ |
| 17 | 6 | 17 | $-28/17$ | $(1,6)$ | $-2$ | $a_{53} = 1214988222$ | $+1$ |
| 18 | 6 | 19 | $-32/19$ | $(0,4)$ | $-5$ | $a_{14} = 4305$ | $+1$ |
| 19 | 6 | 19 | $-32/19$ | $(0,5)$ | $-11$ | $a_{48} = 246563262$ | $+1$ |
| 20 | 6 | 23 | $-40/23$ | $(0,4)$ | $-1$ | $a_5 = 33$ | $+1$ |
| 21 | 6 | 23 | $-40/23$ | $(0,5)$ | $-7$ | $a_{26} = 817329$ | $+1$ |
| 22 | 6 | 23 | $-40/23$ | $(0,6)$ | $-13$ | $a_{69} = 53152478501$ | $+1$ |
| 23 | 6 | 23 | $-40/23$ | $(1,8)$ | $-2$ | $a_{81} = 99284868952$ | $+1$ |
| 1 | 7 | 3 | $1/3$ | $(0,1)$ | $-4$ | $a_3 = 2$ | $+1$ |
| 2 | 7 | 3 | $1/3$ | $(1,1)$ | $-1$ | $a_2 = 4$ | $+1$ |
| 3 | 7 | 4 | $-1/4$ | $(0,1)$ | $-3$ | $a_1 = 2$ | $+1$ |
| 4 | 7 | 5 | $-3/5$ | $(0,1)$ | $-2$ | $a_1 = 6$ | $+1$ |
| 5 | 7 | 5 | $-3/5$ | $(0,2)$ | $-9$ | $a_{29} = 660901$ | $+1$ |
| 6 | 7 | 5 | $-3/5$ | $(1,2)$ | $-4$ | $a_{20} = 1764$ | $+1$ |
| 7 | 7 | 6 | $-5/6$ | $(0,1)$ | $-1$ | $a_1 = 10$ | $+1$ |
| 8 | 7 | 6 | $-5/6$ | $(0,2)$ | $-8$ | $a_{18} = 2288$ | $+1$ |
| 9 | 7 | 6 | $-5/6$ | $(1,2)$ | $-2$ | $a_{10} = 466$ | $+1$ |
| 10 | 7 | 8 | $-9/8$ | $(0,2)$ | $-6$ | $a_8 = 2$ | $+1$ |
| 11 | 7 | 8 | $-9/8$ | $(0,3)$ | $-13$ | $a_{60} = 7895499881$ | $+1$ |
| 12 | 7 | 8 | $-9/8$ | $(1,3)$ | $-5$ | $a_{39} = 26356888$ | $+1$ |
| 13 | 7 | 9 | $-11/9$ | $(0,2)$ | $-5$ | $a_6 = 68$ | $+1$ |
| 14 | 7 | 9 | $-11/9$ | $(0,3)$ | $-12$ | $a_{45} = 117148224$ | $+1$ |

Table 6: Quasi-characters.

| S. No. | $p$ | $u$ | $m$ | $(n, k)$ | $a_0$ | $a_s$ | $\text{sign}(a_{90})$ |
|---|---|---|---|---|---|---|---|
| 15 | 7 | 9 | $-11/9$ | $(1, 3)$ | $-3$ | $a_{24} = 280605$ | $+1$ |
| 16 | 7 | 9 | $-11/9$ | $(2, 4)$ | $-1$ | $a_{43} = 12703782$ | $+1$ |
| 17 | 7 | 10 | $-13/10$ | $(0, 2)$ | $-4$ | $a_4 = 12$ | $+1$ |
| 18 | 7 | 10 | $-13/10$ | $(0, 3)$ | $-11$ | $a_{35} = 12260214$ | $+1$ |
| 19 | 7 | 10 | $-13/10$ | $(1, 3)$ | $-1$ | $a_{12} = 1193$ | $+1$ |
| 20 | 7 | 10 | $-13/10$ | $(1, 4)$ | $-8$ | $a_2 = 1297242310936$ | $+1$ |
| 1 | 8 | 3 | $2/3$ | $(0, 1)$ | $-5$ | $a_5 = 21$ | $+1$ |
| 2 | 8 | 3 | $2/3$ | $(1, 1)$ | $-2$ | $a_4 = 24$ | $+1$ |
| 3 | 8 | 5 | $-2/5$ | $(0, 1)$ | $-3$ | $a_1 = 4$ | $+1$ |
| 4 | 8 | 5 | $-2/5$ | $(0, 2)$ | $-11$ | $a_{40} = 31317813$ | $+1$ |
| 5 | 8 | 5 | $-2/5$ | $(1, 2)$ | $-6$ | $a_{31} = 168402$ | $+1$ |
| 6 | 8 | 5 | $-2/5$ | $(2, 2)$ | $-1$ | $a_{13} = 90$ | $+1$ |
| 7 | 8 | 7 | $-6/7$ | $(0, 1)$ | $-1$ | $a_1 = 12$ | $+1$ |
| 8 | 8 | 7 | $-6/7$ | $(0, 2)$ | $-9$ | $a_{17} = 425$ | $+1$ |
| 9 | 8 | 7 | $-6/7$ | $(1, 2)$ | $-2$ | $a_9 = 282$ | $+1$ |
| 10 | 8 | 7 | $-6/7$ | $(1, 3)$ | $-10$ | $a_{94} = 13594626504306$ | $+1$ |
| 11 | 8 | 7 | $-6/7$ | $(2, 3)$ | $-3$ | $a_{51} = 394666608$ | $+1$ |
| 12 | 8 | 9 | $-10/9$ | $(0, 2)$ | $-7$ | $a_9 = 372$ | $+1$ |
| 13 | 8 | 9 | $-10/9$ | $(0, 3)$ | $-15$ | $a_{62} = 3795042033$ | $+1$ |
| 14 | 8 | 9 | $-10/9$ | $(1, 3)$ | $-6$ | $a_{41} = 29850606$ | $+1$ |
| 15 | 8 | 11 | $-14/11$ | $(0, 2)$ | $-5$ | $a_5 = 54$ | $+1$ |
| 16 | 8 | 11 | $-14/11$ | $(0, 3)$ | $-13$ | $a_{38} = 13804479$ | $+1$ |
| 17 | 8 | 11 | $-14/11$ | $(1, 3)$ | $-2$ | $a_{16} = 6996$ | $+1$ |
| 18 | 8 | 13 | $-18/13$ | $(0, 2)$ | $-3$ | $a_2 = 2$ | $+1$ |
| 19 | 8 | 13 | $-18/13$ | $(0, 3)$ | $-11$ | $a_{25} = 500376$ | $+1$ |
| 20 | 8 | 13 | $-18/13$ | $(0, 4)$ | $-19$ | $a_{95} = 33887464629093$ | $+1$ |
| 21 | 8 | 13 | $-18/13$ | $(1, 4)$ | $-6$ | $a_{55} = 898776774$ | $+1$ |
| 22 | 8 | 13 | $-18/13$ | $(2, 5)$ | $-1$ | $a_{56} = 876492702$ | $+1$ |
| 1 | 9 | 4 | $1/4$ | $(0, 1)$ | $-5$ | $a_3 = 7$ | $+1$ |
| 2 | 9 | 4 | $1/4$ | $(1, 1)$ | $-1$ | $a_2 = 8$ | $+1$ |
| 3 | 9 | 5 | $-1/5$ | $(0, 1)$ | $-4$ | $a_1 = 2$ | $+1$ |
| 4 | 9 | 5 | $-1/5$ | $(0, 2)$ | $-13$ | $a_{52} = 517821576$ | $+1$ |
| 5 | 9 | 5 | $-1/5$ | $(1, 2)$ | $-8$ | $a_{44} = 159785964$ | $+1$ |
| 6 | 9 | 5 | $-1/5$ | $(2, 2)$ | $-3$ | $a_{27} = 344835$ | $+1$ |
| 7 | 9 | 7 | $-5/7$ | $(0, 1)$ | $-2$ | $a_1 = 10$ | $+1$ |
| 8 | 9 | 7 | $-5/7$ | $(0, 2)$ | $-11$ | $a_{23} = 14469$ | $+1$ |
| 9 | 9 | 7 | $-5/7$ | $(1, 2)$ | $-4$ | $a_{15} = 4844$ | $+1$ |
| 10 | 9 | 7 | $-5/7$ | $(2, 3)$ | $-6$ | $a_{88} = 2597931389088$ | $+1$ |
| 11 | 9 | 8 | $-7/8$ | $(0, 1)$ | $-1$ | $a_1 = 14$ | $+1$ |

Table 6: Quasi-characters.

| S. No. | $p$ | $u$ | $m$ | $(n,k)$ | $a_0$ | $a_s$ | sign($a_{90}$) |
|---|---|---|---|---|---|---|---|
| 12 | 9 | 8 | $-7/8$ | $(0,2)$ | $-10$ | $a_{17} = 16286$ | $+1$ |
| 13 | 9 | 8 | $-7/8$ | $(1,2)$ | $-2$ | $a_8 = 114$ | $+1$ |
| 14 | 9 | 8 | $-7/8$ | $(1,3)$ | $-11$ | $a_{86} = 2892946308046$ | $+1$ |
| 15 | 9 | 8 | $-7/8$ | $(2,3)$ | $-3$ | $a_{45} = 32005560$ | $+1$ |
| 16 | 9 | 10 | $-11/10$ | $(0,2)$ | $-8$ | $a_9 = 180$ | $+1$ |
| 17 | 9 | 10 | $-11/10$ | $(0,3)$ | $-17$ | $a_{64} = 27582873039$ | $+1$ |
| 18 | 9 | 10 | $-11/10$ | $(1,3)$ | $-7$ | $a_{43} = 83907339$ | $+1$ |
| 19 | 9 | 11 | $-13/11$ | $(0,2)$ | $-7$ | $a_7 = 129$ | $+1$ |
| 20 | 9 | 11 | $-13/11$ | $(0,3)$ | $-16$ | $a_{51} = 948409950$ | $+1$ |
| 21 | 9 | 11 | $-13/11$ | $(1,3)$ | $-5$ | $a_{30} = 2409958$ | $+1$ |
| 1 | 10 | 3 | $4/3$ | $(0,1)$ | $-7$ | $a_9 = 177$ | $+1$ |
| 2 | 10 | 3 | $4/3$ | $(1,1)$ | $-4$ | $a_7 = 12$ | $+1$ |
| 3 | 10 | 3 | $4/3$ | $(2,1)$ | $-1$ | $a_4 = 6$ | $+1$ |
| 4 | 10 | 7 | $-4/7$ | $(0,1)$ | $-3$ | $a_1 = 8$ | $+1$ |
| 5 | 10 | 7 | $-4/7$ | $(0,2)$ | $-13$ | $a_{30} = 734574$ | $+1$ |
| 6 | 10 | 7 | $-4/7$ | $(1,2)$ | $-6$ | $a_{22} = 123386$ | $+1$ |
| 7 | 10 | 7 | $-4/7$ | $(3,3)$ | $-2$ | $a_{60} = 5880640588$ | $+1$ |
| 8 | 10 | 9 | $-8/9$ | $(0,1)$ | $-1$ | $a_1 = 16$ | $+1$ |
| 9 | 10 | 9 | $-8/9$ | $(0,2)$ | $-11$ | $a_{16} = 4005$ | $+1$ |
| 10 | 10 | 9 | $-8/9$ | $(1,2)$ | $-2$ | $a_8 = 318$ | $+1$ |
| 11 | 10 | 9 | $-8/9$ | $(1,3)$ | $-12$ | $a_{83} = 2147938578036$ | $+1$ |
| 12 | 10 | 9 | $-8/9$ | $(2,3)$ | $-3$ | $a_{42} = 46836198$ | $+1$ |
| 13 | 10 | 11 | $-12/11$ | $(0,2)$ | $-9$ | $a_{10} = 1350$ | $+1$ |
| 14 | 10 | 11 | $-12/11$ | $(0,3)$ | $-19$ | $a_{65} = 9147426057$ | $+1$ |
| 15 | 10 | 11 | $-12/11$ | $(1,3)$ | $-8$ | $a_{45} = 330185968$ | $+1$ |
| 16 | 10 | 13 | $-16/13$ | $(0,2)$ | $-7$ | $a_6 = 136$ | $+1$ |
| 17 | 10 | 13 | $-16/13$ | $(0,3)$ | $-17$ | $a_{44} = 183378033$ | $+1$ |
| 18 | 10 | 13 | $-16/13$ | $(1,3)$ | $-4$ | $a_{22} = 50592$ | $+1$ |
| 19 | 10 | 13 | $-16/13$ | $(2,4)$ | $-1$ | $a_{36} = 7004477$ | $+1$ |
| 20 | 10 | 17 | $-24/17$ | $(0,2)$ | $-3$ | $a_2 = 10$ | $+1$ |
| 21 | 10 | 17 | $-24/17$ | $(0,3)$ | $-13$ | $a_{22} = 179790$ | $+1$ |
| 22 | 10 | 17 | $-24/17$ | $(0,4)$ | $-23$ | $a_{85} = 2870243892555$ | $+1$ |

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
