# Peer review of "Quasi-Characters in $\widehat{su(2)}$ Current Algebra at Fractional Levels"

_SciPost Physics Core, doi:SciPost Phys. Core 6, 068 (2023)_

## Round 1 · Referee Report · Anonymous (Referee 1) · 2023-8-17

Report

I am happy to recommend publication in SciPost Physics Core. There are two optional corrections I noticed when reading the revised version.

Requested changes

  1. After (1.1), there is a statement about the highest weight characters being THE linear independent ones in the extended (ie including relaxed) category. This should be clarified because it is true that the hw chars are linearly independent, but they are far from being the only ones, ie they do not form a basis of characters for the extended category.

  2. Before (2.19), the terms odd (-) and even (+) are explained, but I think the - and + needs to be swapped, cf (2.25).

---

## Round 1 · Author Response

Dear editor and referee,

We thank you for the detailed review, which helped improve the quality of the manuscript. We have reviewed all points raised by the referee in their report and made changes accordingly.

As suggested by the editor, we have decided to resubmit the revised manuscript to SciPost Physics Core instead of the original journal SciPost Physics.

Thank you very much.

Sincerely,
Sachin.

---

## Round 1 · List of Changes

List of changes:

In the following, we mention the comment by the referee, followed by our response.

--0.Exposition of even characters: "I would appreciate it if a characterisation similar to this ($\widehat{su(2)}_{-1/2}$), suitably generalised to cover the other admissible levels perhaps, could also be added to the text. My guess is that other readers would appreciate this kind of remark."
Changes: We improved section 2.2 on even characters by adding the example of $\widehat{su(2)}_{-1/2}$ and a generalisation of the argument presented in this example, which leads to the $q$-expansion of the even character.

--1. "The end of the second paragraph suggests that CFTs with a finite number of primaries are rational. This is false and the original counterexample is the triplet model, see hep-th/9604026."
Changes: We amended the introduction's second paragraph to clarify this point.

--2. "The word "representation" appears to be used frequently to mean "irreducible highest-weight representation" or something similar (but perhaps not always). Perhaps it could be clarified that this is the case unless otherwise noted?"
Changes: We added a footnote in the introduction before eqn (1.1) to the same effect.

--3."Footnote 3 claims that "admissible" will be defined in the next section. But, I didn't see it defined there (or anywhere else). Is it the Kac-Wakimoto definition from their 1988 paper, so only applicable to irreducible highest-weight representations? Or does it mean any representation of the vertex operator algebra underlying the CFT under consideration? Or something else?"
Changes: We define the admissible (unflavoured) characters which satisfy the properties mentioned at the beginning of section 2 in the manuscript. To state this clearly, we have rephrased the paragraph below eqn (2.1) and the footnote (now footnote 5) so as to not confuse with admissibility of a representation.

--4."The term "big category of admissible modules" after (1.1) is perplexing, especially as reference is made to [17] where the term is never used. Please be more precise here."
Changes: We removed "big category" and added "extended category of admissible modules, which includes the relaxed category along with the indecomposables" to the paragraph after eqn (1.1). We also replaced the reference [17] with [11,13], where this is described.

--5."There appears to be a systematic misunderstanding of the term "integrable" in the paper. As far as I can tell, it is universally understood (in this context) to mean that a representation decomposes as a direct sum of finite dimensional $su(2)$ representations, for any choice of $su(2)$ subalgebra. As such, $SU(2)_k$ has no (non-zero) integrable highest weight representations unless k is a non-negative integer. However, the author mentions them in many places for fractional levels. Perhaps they are thinking of representations whose characters converge at $z=0$, ie. finite-dimensionality for one special choice of $su(2)$ subalgebra?"
Changes: We removed the word integrable from several places where a wrong meaning was evident. In the remaining places, the integrable representations only apply to the irreducible highest weight representations of $\widehat{su(2)}_k$, $k\in\mathbb{Z}_{>0}$.

--6." In the second paragraph of Sec. 1.1, what does "admissibility of the even characters" mean? These objects don't correspond to representations in general. Do you mean both the representations whose characters are being subtracted are admissible?"
Changes: The admissibility of even unflavoured characters borrows the definition of admissible characters we used in section 2 of the manuscript. We clarified further by adding a statement in the second paragraph of section 1.1.

--7."After (2.4), the RCFT discussion suggests that we add the constraint of positivity of fusion coefficients if the S-matrix is unitary. But, the S-matrix of a RCFT is always unitary... "
Author response: The S-matrix, as defined in eqn (2.4) of the manuscript, is not necessarily unitary. In particular, since the diagonal partition function of eqn (2.3) is modular invariant, we have $S^{\dagger}MS=M$ where we are not restricting $M$ to be the identity matrix. If $M$ is the identity matrix, we see the relation $S^{\dagger}S=I$.

--8."Immediately after, I didn't understand what was meant by "reduced S and T-modular matrices"."
Changes: We borrowed the name reduced S-matrix from reference [29] of the revised manuscript. The reduced S and T-matrices act on the unflavoured characters. We clarify this by rephrasing the paragraph below eqn (2.4).

--9."I also didn't understand "a unitary S-matrix implies a 1-1 correspondence of the unflavoured characters with the modules". What could unitarity possibly have to do with the linear independence of $q$-characters?"
Author response: If the S-matrix is not unitary, it does not preserve (under conjugation) the identity matrix. Since it preserves another diagonal matrix M, which has at least one entry greater than one and an integer, it implies more than one representation has the same unflavoured character.

--10. "After (2.7), it is claimed that the spectrum is invariant under $j\to -1-j$. But, this is false as (2.9) notes."
Changes: We only mean the energy spectrum ($L_0$ eigenvalues). We have clarified this statement in the manuscript.

--11. "The last sentence in Sec. 2.1 is baffling, especially as it references [17]. Shouldn't a $q$-expansion (about $q=0$) always give the degeneracies (multiplicities) of weights in a module? The subtle discussion in [17] concerns expansions in $\omega$ (or $z$) which is much trickier."
Changes: We understand that the last line in section 2.1 needs to be clarified. However, we have removed this sentence since it did not contribute anything new.

--12. "Is there a typo in (2.20)?"
Changes: Yes, there was a typo in the equation. We have corrected the equations (2.34) and the equation (2.19) in the revised manuscript.

--13. "I guess (2.23) only holds for $k\neq0$?"
Changes: True, it only holds for $k\neq 0$. We added this to the left-hand side of equation (2.37) in the revised manuscript.

--14. "I also dislike the reference in the first paragraph of Sec. 2.3 to [17], seemingly in support of something "manifest" that I honestly find incomprehensible."
Changes: Following the comment, we rephrased the first paragraph of section 2.3.

--15. "In Sec. 3.1, it may be worth noting that the term "threshold level" also has the name "boundary level" in the literature. I believe it was introduced by Kac-Roan-Wakimoto in their 2003 article, but it could have been earlier."
Changes: We included this comment and the reference in section 3.1.

--16." I'll add that it is somewhat dangerous to identify a unitary CFT from its modular data. There are many known examples where different RCFTs give the same representation of the modular group. I would not be surprised if there were also examples of inequivalent RCFTs whose q-characters matched!"
Changes: We admit that the modular data alone is insufficient to characterise the CFT fully. We have added a statement in the conclusions at the end of the first paragraph to notify the reader of our shortcomings.

--17."In the conclusion, it is claimed that even characters are expanded in the region $|z|<1$ and $|q|<1$ and that the radius of convergence is the same as that for a RCFT. However, this can't be correct as there can be poles in $|z|<1$ depending on $\tau$, right?"
Changes: Following the discussion with the referee, we correct the radius of convergence to only $|q|<1$. We changed the relevant part in the second paragraph of the conclusions.

---

## Editorial Decision

published